# Intercomparison of different uncertainty sources in hydrological climate change projections for an alpine catchment (upper Clutha River, New Zealand)

Andreas M. Jobst[1], Daniel G. Kingston[1], Nicolas J. Cullen[1] and Josef Schmid[2]

[1]Department of Geography, University of Otago, Dunedin, PO Box 56, New Zealand
[2]Department of Geography, University of Munich (LMU), Munich, Germany

*Correspondence to*: Andreas M. Jobst (andreas.jobst@otago.ac.nz)

**Abstract.** As climate change is projected to alter both temperature and precipitation, snow controlled mid-latitude catchments are expected to experience substantial shifts in their seasonal regime, which will have direct implications for water management. In order to provide authoritative projections of climate change impacts, the uncertainty inherent to all components of the modelling chain needs to be accounted for. This study assesses the uncertainty in potential impacts of climate change on the hydro-climate of a headwater sub-catchment of New Zealand's largest catchment (the Clutha River) using a fully distributed hydrological model (WaSiM) and unique ensemble encompassing different uncertainty sources: General Circulation Model (GCM), emission scenario, bias correction and snow model. The inclusion of snow models is particularly important, given that (1) they are a rarely considered aspect of uncertainty in hydrological modelling studies, and (2) snow has a considerable influence on seasonal patterns of river flow in alpine catchments such as the Clutha. Projected changes in river flow for the 2050s and 2090s encompass substantial increases in streamflow from May to October, and a decline between December and March. The dominant drivers are changes in the seasonal distribution of precipitation (for the 2090s +29 to +84% in winter) and substantial decreases in the seasonal snow storage due to temperature increase. A quantitative comparison of uncertainty identified GCM structure as the dominant contributor in the seasonal streamflow signal (44-57%) followed by emission scenario (16-49%), bias correction (4-22%) and snow model (3-10%). While these findings suggest that the role of the snow model is comparatively small, its contribution to the overall uncertainty was still found to be noticeable for winter and summer.

## 1 Introduction

Over recent decades climate change has had a considerable impact on the Earth's freshwater resources (Jiménez Cisneros et al., 2014), causing, amongst others, changes in the amount of runoff (Piao et al., 2010), the timing of peak discharge (Hidalgo et al., 2009), a reduction in glacier volume (Rosenzweig et al., 2007) and an increase in flood risk (Pall et al., 2011). Future impacts under mid and late 21[st] century climate change are projected to intensify, affecting both the main processes and stores of the water cycle. The impacts include an increase of potential evapotranspiration (PET) over most land areas, a further

shrinkage of glaciers and changes in the runoff regime of snowmelt affected basins (Jiménez Cisneros et al., 2014). Thus 21[st] century climate change is expected to have substantial implications for water users and operators alike, which makes robust projections of potential changes in the seasonality and magnitude of streamflow essential.

While General Circulation Model (GCM) land surface schemes can be used for climate change impact assessments (e.g. Haddeland et al., 2011; Gudmundsson et al., 2012), their coarse resolution makes them inadequate for modelling studies at the small and meso scale. Thus, climate change impact studies typically use a cascade of models and processing steps to move between the scales of the lower resolution climate models and a separate higher resolution hydrological model (Maraun et al., 2010; Muerth et al., 2013).

As discussed by Muerth et al. (2013), the hydro-climatic model chain typically consists of the following components: emission scenario, GCM, Regional Climate Model (RCM) or statistical downscaling, bias correction and hydrological model. All of these components constitute a potential uncertainty source, and as such all need to be examined to provide a truly comprehensive understanding of the uncertainty associated with hydrological impact assessments (Teutschbein and Seibert, 2010). The uncertainty associated with the individual components of the model chain has been investigated by an increasing number of studies. Typically, GCM structure is identified as the dominant source of uncertainty (e.g. Graham et al., 2007; Prudhomme and Davies, 2009; Hagemann et al., 2011; Dobler et al., 2012). There is little agreement on the second most important source of uncertainty between the downscaling method (Wilby and Harris, 2006; Prudhomme and Davies, 2009; Dobler et al., 2012), the bias correction (Vormoor et al., 2015) or the emission scenario (Bennett et al., 2012). A common finding is that hydrological model uncertainty is less important than other uncertainty sources (i.e. GCM), but cannot be ignored (Prudhomme and Davies, 2009; Teng et al., 2012; Thompson et al., 2013; Velázquez et al., 2013). However, for certain hydrological indicators (e.g. high flow events) hydrological models can be associated with a comparable uncertainty range to the driving climate projections (e.g. Ludwig et al., 2009; Muerth et al., 2012).

As an alternative to an ensemble of different hydrological models varying in their representation of spatial variation (i.e. lumped, semi-distributed, fully distributed) and process descriptions (i.e. stochastic, conceptual or physically oriented), some studies have explored uncertainty associated with particular routines within a single model. Examples include the sensitivity of climate change impacts on the PET method used (e.g. Kay and Davies, 2008; Thompson et al., 2014). However, in snowmelt affected mid latitude catchments PET-related uncertainty is often relatively small (e.g. Koedyk and Kingston, 2016), with uncertainty linked to snow-related processes more important. For example, Troin et al. (2016) investigated the uncertainty introduced by the snowmelt routine in a hydrological model for three Canadian catchments. For a number of snow indicators (e.g. snow water equivalent (SWE)), most of the uncertainty was found to be caused by natural climate variability. For temporal indices (e.g. duration of snow pack) however, the different snow models showed a greater variability. Troin et al. (2016) did not look at the implications of snow model uncertainty for river flow, but the greater uncertainty associated with temporal indices could be indicative of significant implications on the timing of snowmelt and so the annual streamflow regime. Thus, the choice of the snow model as a potential uncertainty source and its implications on streamflow needs to be explored further, particularly in alpine catchments.

The aim of this present study is to investigate the contribution of the snow model and three more commonly studied uncertainty sources (i.e. GCM, emission scenario and bias correction method) to the climate change signal in hydrological projections. New Zealand's largest catchment, the Clutha, was selected for this purpose as its highly complex hydro-climate, including snow affected headwaters, makes it a particularly interesting case study. To this end, the fully distributed hydrological model

WaSiM (Schulla, 2012) was implemented for an alpine sub-catchment of the Clutha, with a total of 32 separate hydrological simulations produced. These comprised two emissions scenarios, four GCMs, two bias correction methods and two snow models. Although previous New Zealand studies (including for the Clutha) have examined multiple GCM scenarios (e.g. Poyck et al., 2011; Gawith et al., 2012; Caruso et al., 2016), none have used an ensemble covering the present range of uncertainty sources. Furthermore, in using WaSiM this will be the first application of a fully distributed and grid-based hydrological model

for this purpose in a medium to large-scale New Zealand catchment. Consequently, this study will generate the most complete assessment of climate change impacts on streamflow and associated uncertainty for an alpine New Zealand catchment. Importantly, the study will also speak more widely to the issue of snowmelt uncertainty under climate change in alpine catchments.

## 2 Data and methods

### 2.1 The study area

The Clutha/Mata-Au is the largest catchment (20586 km$^2$) in New Zealand and is situated in the lower half of the South Island, extending eastwards from the Southern Alps (Figure 1a). It has the highest average streamflow of any river in New Zealand (approximately 570 m$^3$ s$^{-1}$) and drains 6% of the South Island's water (Murray, 1975). The catchment is characterized by a highly variable hydro-climate ranging from very humid alpine headwaters dominated by seasonal snow accumulation and

melt, to substantially drier areas in the central catchment. The Clutha catchment can thus be considered broadly representative of most of the South Island's hydrologic and climatic domain, and so an ideal candidate for investigating climate change impacts.

As described in Jobst (2017) WaSiM was implemented for the entire Clutha catchment as a tool for climate change impact modelling. Most of the upper and lower Clutha catchment are under extensive water management (particularly for hydro-

electric dams and water abstractions), except for the north-western part (gauge Chards Rd in Figure 1b), which is characterised by natural flow conditions. As the focus of this climate change impact study is on potential changes in natural streamflow and seasonal snow, the results focus mostly on the Kawarau River sub-catchment (see Jobst (2017) for model simulations at the other key sites of the Clutha). The catchment area of the Kawarau River (at Chards Rd) is 4541 km$^2$ (22% of entire Clutha basin) and with a mean discharge of 212 m$^3$ s$^{-1}$ is the largest component of the upper Clutha basin, comprising 36% of flow at

the catchment outlet. Snowmelt contributes approximately 20% of annual flow. Streamflow at Chards Rd is also highly correlated with the headwater sub-catchment of Lake Wanaka (located to the east of the Kawarau sub-catchment), while the outflow of Lake Hawea is controlled by a dam for hydropower generation further downstream.

Most of the Kawarau sub-catchment is covered by indigenous tussock grassland followed by low producing exotic grassland and forest (Figure 1c). The elevation of the sub-catchment ranges between 300 and 2800 m. The ice-covered area inside the Kawarau catchment is small and based on 2001/2002 satellite imagery amounts to 1.8% or 84 km$^2$ (New Zealand's Land Cover Database v3.0 as published by Landcare Research in 2012). Chinn (2001) estimated the volume of ice in the Kawarau basin at 2.25 km$^3$ using the area mean-depth relationship of the World Glacier Monitoring Service. Thus, based on Chinn's (2001) estimate the water stored in the glaciers of the Kawarau could only sustain its mean flow for ~123 days, which highlights that glaciers in the catchment are less important from a hydrological perspective.

## 2.2 The WaSiM model of the Clutha

The fully distributed and physically-oriented hydrological model WaSiM-Richards (version 9.06.10) was implemented at a spatial resolution of 1 km and at a daily time step. The main components of this implementation of WaSiM are described briefly here – for a more detailed description see Schulla (2012). The modelling of PET is solved by the Penman Monteith approach, while actual evapotranspiration (ET) is a function of the simulated soil water content. Soil and groundwater processes are described by finite differencing of the 1D-Richards equation combined with a 2D groundwater model. In addition, WaSiM's dynamic glacier model was used to describe the glacial processes for the ice-covered cells located in the upper catchment. The glacier model uses three degree-day factors to calculate melt from the three storage components snow, firn and ice. After each year the remaining snow of a cell forms a new firn layer and once the firn stack reaches seven layers (as recommended by Schulla (2012)) the lowest firn layer turns into ice.

WaSiM was parameterised using both remotely sensed data (i.e. MODIS-15A2-1km for leaf area index) and values obtained from the literature. Two versions of this WaSiM implementation were set up, one with a simple temperature index (Tindex) snowmelt routine (Schulla, 2012) and the other with the conceptual energy balance model of Anderson (1973). The Tindex model calculates the melting rate via a degree-day factor multiplied by the difference between actual temperature and the melting point temperature. The Anderson model is more complex as it computes four separate melt fractions, accounts for radiation by using a seasonal melt factor and also models the refreezing of liquid water stored in the snow pack when the actual temperature is below melting point.

Station-based meteorological observations of mean daily air temperature (Tmean), precipitation, solar radiation, relative humidity and wind speed were interpolated (Jobst, 2017; Jobst et al., 2017) and served as input to WaSiM during the calibration (2008-2012) and validation (1992-2008) periods. The last four hydrological years of the reference period were chosen for calibration because of the higher density of weather stations compared to previous years and a better consistency of the streamflow records. Further the relatively short calibration period constitutes a compromise between a reasonable processing time and a sufficient number of iterations (as part of the auto-calibration). Although the calibration period is rather short this means that the longer validation period allows for a robust assessment of the hydrological model's performance.

After a two-year model spin-up the individual sub-models of WaSiM (unsaturated zone, groundwater, snow and glacier model) were calibrated iteratively using a combination of auto-calibration and manual parameter optimization (Figure 2; see Jobst

(2017) for a more detailed description of the calibration process). Particle swarm optimization (PSO) (Kennedy and Eberhart, 1995) was used for auto-calibration due to its effective performance during the first iterations and fast operation (Jiang et al., 2010), allowing for an adequate compromise between processing time and efficiency. First the parameters controlling subsurface flow were calibrated using the Nash-Sutcliffe model coefficient of efficiency (NSE). Therefore, the NSE was based

on daily logarithmic streamflow values ($NSE_{log}$) to account for the physical link between low flow conditions and subsurface flow components of WaSiM. The remaining parameters, controlling surface flow were calibrated using regular daily streamflow values (standard NSE) to preserve the sensitivity of the NSE to flood peaks. A regionalisation based on spatial proximity and topographical similarity was carried out next (see map in Figure 2) to parameterise sub-catchments that were ungauged or only had short records of streamflow (e.g. parameters of sub-catchment 4 → sub-catchment 3).

In the following step the two snow models were calibrated for three separate headwater sub-catchments (gauges: The Hillocks, Peat's Hut and West Wanaka as shown in Figure 1b) against monthly streamflow ($NSE_{mo}$). The resulting parameter sets were then averaged resulting in a global parameter set for each of the two snow models respectively (Table 1).

Opposed to the other sub-models the glacier model was calibrated manually for the entire Clutha catchment using the annual volume estimates (1994-2010) of Willsman (2011) as calibration (1994-2001) and validation (2002-2010) source. The degree

day factor (DDF) of ice (7.17 mm$°C^{-1}d^{-1}$) was based on the study of Anderson et al. (2006), while the DDF of snow (3.80 mm$°C^{-1}d^{-1}$) was calibrated manually. The DDF of firn (5.54 mm$°C^{-1}d^{-1}$) was calculated by averaging the DDFs of snow and ice, to ensure a physically sound relationship between the three parameters (DDF of ice > DDF of firn > DDF of snow).

With regards to the validation of streamflow daily NSE values indicated a reasonable performance for the three tributaries (Dart, Shotover and Matukituki River), while monthly NSE values indicated a good performance at these sites (Table 2). For

the Matukituki River the validation of both WaSiM versions revealed a substantially better simulation of monthly streamflow (NSE of 0.83 and 0.82, respectively) when compared to the TopNet based modelling study of Gawith et al. (2012) (NSE of 0.68). For Chards Rd the validation of WaSiM-Anderson and WaSiM-Tindex revealed a strong performance at the daily and monthly time scale, with NSE values between 0.85 and 0.90 across all model versions, timescales and time periods (Table 2). The monthly hydrographs of WaSiM-Anderson and WaSiM-Tindex (Figure 4c) further indicate a realistic representation of

observed daily runoff at Chards Rd. Obvious inaccuracies of both WaSiM versions are an underestimation of larger flow events during the melt period (e.g. -January 1995, 1996 and 2000) and an overestimation during autumn (e.g. April-May 1994 and 1996). The likeliest explanation is that not enough snow is being accumulated from autumn to early winter and consequently the main melt peaks are under-simulated. This is largely supported by the monthly SWE simulations (Figure 4b) and can be exemplified for the snow accumulation period of 1992 where WaSiM-Anderson results in a greater accumulation of SWE

compared to WaSiM-Tindex. In the following melt period (November 1992 to February 1993) the observed peak is then approximated better by WaSiM-Anderson due to the larger SWE storage. Overall inaccuracies are generally larger with WaSiM-Tindex as can be seen in December 1994 (Figure 4). Compared to WaSiM-Anderson the snow pack melts slower which causes a greater underestimation of runoff during the main melt period. Overall the good performance of WaSiM is

substantiated by the monthly NSE values which correspond to 0.89 and 0.88 for WaSiM-Anderson and WaSiM-Tindex, respectively.

The modelled glacier storage decreased from 540 mm to 265 mm during the reference period. For comparison Willsman (2011) found a decrease of 29% for the entire Clutha catchment between 1994 and 2010, which is less than the decrease modelled here for the Kawarau (42%). As there are no other studies that have quantified the glacier volume of the Kawarau the results of this study couldn't be assessed any further.

Due to the high annual precipitation totals the grid cells in the upper catchment could potentially be prone to large build-ups of snow that persist throughout the melting period and into the next hydrological year. As discussed by Freudiger et al. (2017) such snow towers can cause substantial errors in the modelled water balance and should therefore be made transparent. As indicated in Figure 4b the snow pack of some cells persists across all of the melt seasons with SWE values at the end of the melt season ranging between 7 and 46 mm (21 and 77 mm) for WaSiM-Anderson (WaSiM-Tindex). To further investigate the significance of this potential issue, all cells where the SWE value at the end of the hydrological year exceeded the annual snowfall (excluding glacier cells), were marked. The combined relative area of these cells is shown in Figure 4a and does not exceed 1% for WaSiM-Anderson, while for WaSiM-Tindex the affected area does not exceed 2% except for the years 1997 and 1998. The mean exceedance (SWE/snowfall) for the individual years based on all affected cells varies from 1.3 to 1.8 for WaSiM-Anderson and from 1.4 to 2.3 for WaSiM-Tindex. Thus, for a number of cells along the main divide the modelled build-ups of snow could be classified as potential snow towers. However, the combined area of these cells is small, which means that the effect on the modelled water balance is negligible.

## 2.3 The model cascade

Most existing impact studies in the New Zealand domain (Poyck et al., 2011; Srinivasan et al., 2011; Zammit and Woods, 2011; Zemansky et al., 2012) have been based on statistically downscaled GCM simulations provided by the National Institute of Water and Atmospheric Research (NIWA) (Ministry for the Environment, 2008). More recently a small ensemble of four GCMs (CM2.1-GFDL, ECHAM5, HadCM3 and MK3.5-CSIRO) based on the A1B and A2 SRES emissions scenarios has been dynamically downscaled for the New Zealand domain using the HadRM3P RCM (Ackerley et al., 2012), and it is this ensemble of eight dynamically downscaled GCM simulations that forms the data set for the current study.

A model chain was constructed (Figure 3) to process the raw RCM runs (from 1990 to 2099) and generate high resolution climate change projections at the hydrological model scale. Two different bias correction methods, linear transformation (LT; as described in Lenderink et al. (2007)) and quantile mapping (QM; as described in Mpelasoka and Chiew (2009)), were used to correct the RCM data. Both methods have been successfully used by a number of studies (e.g. Boé et al., 2007; Chen et al., 2013; Gutjahr and Heinemann, 2013) and were selected here to allow for a direct comparison between a simple correction method based on additive or multiplicative correction terms (LT) and the more complex distribution-based QM approach.

To bridge the gap between the RCM grid (~27 km) and the hydrological model grid (1 km) an additional statistical downscaling step was required. The downscaling of precipitation (and the remaining three variables) is based on the topographical scaling

approach of Frueh et al. (2006), while maximum and minimum air temperature are scaled via monthly lapse rate models (as described in Jobst et al. (2017) but excluding the thin plate spline layer). As part of the downscaling, additional processing steps were adopted from Marke (2011) to ensure the conservation of mass and energy when transforming the RCM data between the model scales.

## 3. Results

### 3.1 Baseline simulations

For the historic analysis, the ensemble was divided into four sub-ensembles composed of the two bias correction methods and the two snow models (i.e. QM-Anderson, QM-Tindex, LT-Anderson and LT-Tindex). The regimes (i.e. monthly runoff values averaged across all years) of the eight RCM driven simulations of each sub-ensemble were compared to the observed runoff regime (OBS) and the modelled runoff regime forced by the observed meteorology (MOD-METEO$_{OBS}$).

The skill in reproducing the observed historic regime varies substantially depending on both the bias correction method and the snowmelt routine (Figure 5). Overall QM-Anderson gave the most realistic approximation of the observed regime, although still with some overestimation in May (late autumn) followed by an underestimation during July and August (winter). QM-Tindex and LT-Anderson also underestimate the main peak, however the general fit of their RCM members is still relatively close to the observed regime. The largest discrepancies occurred with LT-Tindex, with a substantially flatter regime, mainly due to too much flow being generated between May and September leading to an underestimation of the main peak (November to January). Overall the LT method shows a lower skill in reproducing the observed regime, which is especially pronounced in combination with Tindex. This behaviour points to a high sensitivity of the modelled regime towards the bias correction method and generally speaking the meteorological forcing.

The RCM driven runs agree more closely with MOD-METEO$_{OBS}$ than with the observed regime, as monthly over- and underestimations of MOD-METEO$_{OBS}$ have propagated into the RCM driven WaSiM simulations. This was expected as the RCM climate data have been tuned (i.e. bias corrected) to the station-interpolated meteorology that was used to drive MOD-METEO$_{OBS}$.

Regarding the water balance (Table 3), the observed annual precipitation of the Kawarau sub-catchment (2007 mm) was underestimated by both the QM (1926 mm) and the LT sub-ensemble (1931 mm) during the reference period. A small part of that difference is caused by the shorter 360-day calendar of the RCM runs. Compared to MOD-METEO$_{OBS}$, ET was modelled almost identically by QM-Anderson and QM-Tindex, with slightly larger discrepancies (-1%) under LT-Anderson and LT-Tindex. Regarding runoff, QM and LT resulted in an underestimation of -6 and -5%, respectively, while the choice of the snow model had only a negligible impact.

In terms of the seasonal SWE volume, the QM-Anderson runs agree more closely with MOD-METEO$_{OBS}$ than the other sub-ensembles. The differences in the seasonal SWE volumes range from 3% in spring to 6% in autumn. Compared to MOD-METEO$_{OBS}$ the modelled SWE volumes of LT-Tindex were almost identical for summer and autumn but substantially lower

for winter (-18%) and spring (-16%). Thus, the poor agreement between the observed runoff regime and the LT-Tindex runs (Figure 5) can very likely be explained by too much melt being modelled between winter and early spring. The latter results in a reduced SWE volume, which is insufficient to supply streamflow with enough snowmelt during late spring and summer.

## 3.2 The climate change signals of precipitation and air temperature

The climate change signals of Tmean and precipitation that are presented in the following section correspond to the mean change of the spatio-temporal average between a future period (either 2050s or 2090s) and the reference period. For precipitation, the spread of the 2050s summer climate change signal (Figure 6) is almost completely caused by the GCM structure. Both the emission scenario and bias correction method have negligible effects on the extent of the signal range and median, with the latter showing a near zero change in precipitation. Regarding the 2090s summer, the median change is more

negative, while both the emission scenario and bias correction cause a slight increase in the uncertainty range. A different situation can be seen for the 2050s winter (Figure 6), where the extent of the range is largely determined by the emission scenario. For the 2090s winter, all three components have a considerable impact on the uncertainty range. Here, the GCM spread is the largest of all seasons and future periods. It can also be seen that the precipitation signal is noticeably higher for the A2 sub-ensemble (mainly caused by ECHAM5-A2). In addition, the selection of the bias correction method considerably

increases the extent of the whole ensemble, resulting in a total uncertainty range spanning 55.3 percentage points (i.e. a 28.5% to 83.8% increase from the baseline).

For all seasons the uncertainty in the Tmean signals during the 2050s is predominantly caused by the GCM structure (Figure 7). The selection of the emission scenario becomes a major source of uncertainty in the 2090s with most of the A2 members projecting a stronger signal than their corresponding A1B members. However, this only holds for members stemming from

the same GCM (e.g. ECHAM5-A1B and ECHAM5-A2), as can be seen for the 2090s winter, where an A1B member (MK3.5-CSIRO) has a greater warming signal than two of the A2 members (HadCM3 and CM2.1-GFDL).

## 3.3 The hydrological signals

### 3.3.1 Runoff

For both future periods the historic melt-driven December peak in the annual regime is projected to move earlier in the year

(Figure 8). In the 2050s, the highest monthly mean flow is projected to occur between October and November, with a further shift for the 2090s (to September and October). The most striking transformation for the 2090s is the dramatic enhancement of monthly flows during winter and spring, with uninterrupted increases from May to October.

In order to specifically compare the contribution of the snow model with the remaining sources of uncertainty, the seasonal signals in runoff are shown separately for WaSiM-Anderson and WaSiM-Tindex (Figure 9). It can be seen that the influence

of the snowmelt routine on seasonal flows is comparatively small for both periods and during all seasons. The most noticeable difference is an enhancement of the decrease during summer and a more pronounced increase during winter when using the

Anderson model. Compared to the snow model the effect of the bias correction on the overall spread is more important. Positive signals were found to be enhanced by the QM method and vice versa for negative signals. Further, the influence of bias correction becomes visibly more important in the 2090s (except for autumn).

While both the bias correction and the snow model contribute to the overall spread, the GCM and the choice of the emission scenario appear to be the dominant sources of uncertainty. In most seasons the GCM range differs substantially depending on the underlying emission scenario. In the 2050s period this becomes especially apparent for the autumn season (Figure 9a), during which the A2 runs show a much greater spread than the A1B runs. Regarding the 2090s period (Figure 9b), differences between the A2 and the A1B runs become more pronounced and the most extreme signals are all represented by an A2 member. Regarding the climate change signal derived from the overall ensemble mean, the largest changes were projected for winter and summer. For the 2090s (2050s) summer runoff is projected to decrease by -24% (-10%), with a substantial increase of 71% (29%) during winter. The overall ensemble spread becomes largest for the 2090s winter season, ranging from 40 to 116%.

### 3.3.2 Snow water equivalent

During the historic period simulations of the monthly SWE storage varied considerably for the four sub-ensembles (as expected from the seasonal values in Table 3), with the lowest and highest volumes modelled by LT-Tindex and QM-Anderson, respectively. Despite these differences in the historic simulations, the relative changes are similar between the four sub-ensembles and thus the results are only shown for the QM-Anderson sub-ensemble (Figure 10). A general observation is that the larger spread in the precipitation signal of the QM runs (Figure 6) has clearly propagated into the uncertainty range of the monthly SWE volume, while the LT envelopes are visibly narrower during both time periods (particularly for the 2090s).

For the 2050s period (Figure 10a), the A1B envelope predominantly lies within the upper and lower bounds of the A2 envelope, with the latter showing a substantially greater spread. The proportion of the two envelopes becomes reversed in the 2090s period, when the A1B envelope surpasses the A2 envelope in all months (Figure 10b).

Although the 2090s envelopes of the four sub-ensembles have a relatively large overlapping area during winter and spring, all of the A2 members have a lower SWE volume than their A1B counterparts. It is however noticeable that the A1B member MK3.5-CSIRO has a lower SWE volume than the ECHAM5 A2 member, which is associated with a greater warming signal during winter and spring (Figure 7). A closer inspection of the corresponding climate signals revealed that the MK3.5-CSIRO-A1B signal can be primarily attributed to less precipitation (second smallest increase in winter precipitation of the ensemble), which combined with a relatively strong warming signal (strongest of all A1B members) would have resulted in less snowfall and therefore less snow accumulating. This indicates that despite the warming signal being a key driver of SWE changes in the future, the precipitation signal also plays an important role, which in this case led to an enhancement of the negative SWE signal.

As shown (Figure 11) by the transient simulations of mean SWE from July to December (months with historically the highest SWE) there is no clear distinction between the A1B and A2 simulations until the last quarter of the 21st century when the median of the A2 runs stays consistently under the A1B runs. While the median SWE based on all 16 QM runs decreases

substantially (60%) during the 110-year period, years with large accumulations of SWE still occur in the second half of the 21st century for individual members and years. The negative trend was found to be more pronounced for the glaciers in the catchment, which are projected to lose 93% of their volume (91% based on A1B and 94% based on A2) by 2099.

### 3.4 Quantifying the uncertainty in the seasonal runoff signal

In order to quantify the uncertainty induced by the individual components of the model chain compared to the overall uncertainty in runoff projections, the approach of Muerth et al. (2012) was adopted. First, the approach is exemplified for the uncertainty quantification of the winter runoff signal (Figure 12). An uncertainty component (e.g. GCM) is selected and all possible permutations between the selected and the remaining three model components are computed, resulting in 32 combinations (4 • 2 • 2 • 2). In the next step, the currently selected component is varied (four GCMs = four circles), while the other three components (emission scenario, bias correction and hydrological model) are fixed to a certain combination of their members. As such, all of the four circles spanning the first bar (Figure 12, left, first segment) have the emission scenario fixed to A1B (orange quarter), the bias correction method fixed to QM (white quarter) and the snow model fixed to Anderson (blue quarter), while the fourth quarter, which corresponds to one of the four GCM members (CM2.1-GFDL, ECHAM5, HadCM3 and MK3.5-CSIRO), is varied. Thus, the effect of the GCM has been isolated by fixing the other components to one particular combination. This step is repeated for all the possible combinations between the three remaining components, translating to a total of eight combinations (eight bars). As each of the eight bars in the GCM segment (Figure 12, first segment) contains four circles, all 32 possible permutations have been accounted for. The same procedure is then repeated for the remaining three uncertainty components. The mean contributions to the overall uncertainty (including the standard deviations) of the individual components are then displayed in a radar chart (Figure 13a).

The uncertainty analysis (Figure 13a-d) identified the GCM as the primary source of uncertainty across all seasons (44-57% change in runoff). The selection of the emission scenario was the second largest contributor (16-49%), except for winter when the choice of bias correction was greater (22% vs. 16%). The uncertainty induced by the emission scenario showed a pronounced seasonal variation and was found to be largest during summer (33%) and autumn (49%). A likely explanation for the latter are the significantly different Tmean signals under A1B and A2 (Figure 7), which translate to different ET rates and consequently variable changes in runoff. This is supported by the fact that the most extreme decreases in summer and autumn runoff occurred under the A2 scenario (not shown here). The contribution of bias correction to the overall uncertainty ranged from 4% in autumn to 22% in winter and was higher (except for autumn) than the relative contribution of the snow model (3-10%).

As described in Muerth et al. (2012) the standard deviation associated with the relative uncertainty contribution of an individual component indicates its degree of dependence on the other model components. Here the standard deviation was clearly largest for the emission scenario and the GCM. The standard deviations of both components also varied seasonally and were found to be largest during spring and autumn (Figure 13b, d). Thus, it can be stated that, during spring and autumn the uncertainty induced by the GCM (same holds for emission scenario) is associated with a relatively large dependence on the other variables.

## 4. Discussion

Before the climate change uncertainty assessment was carried out, the hydrological simulations were analysed during the reference period. Performance in reproducing the observed regime varied depending on both the selection of the snow model and the bias correction method. The bias correction method was expected to only have a minor effect on the simulated monthly runoff during the reference period, which made the observed sensitivity somewhat unexpected. A potential explanation could be related to the predominant air temperature during mature precipitation events along the main divide. At Brewster Glacier (located just outside the Clutha catchment, along the main divide west of Lake Hawea – Figure 1), Cullen and Conway (2015) found air temperature to be frequently around the rain-snow threshold during events with major solid precipitation, which led to the conclusion that the accumulation of snow in areas along the main divide is vulnerable to small variations in air temperature. The relatively large variability during the reference period was therefore seen as a first indicator for a potentially high sensitivity of the modelled snow storage and runoff to projected warming. The historic analysis also showed that the observed regime was captured more realistically by WaSiM-Anderson opposed to WaSiM-Tindex. Studies targeting the controlling processes of snowmelt in the Southern Alps (Prowse and Owens, 1982; Sims and Orwin, 2011) identified net radiation as an important driver of snowmelt (in addition to sensible heat). Thus, the better performance of the conceptual energy balance method (WaSiM-Anderson) compared to the Tindex model can likely be explained by the advances of accounting for individual melt fractions and using a seasonal radiation melt factor (see Anderson (1973) or Schulla (2012)).

Overall the baseline analysis showed that the individual sub-ensembles performed differently and that the observed regime was not always enveloped by the corresponding range of simulations. This introduces some additional uncertainty into the projections that could not be quantified or accounted for in this study. A potential explanation is that neither snow model was able to accurately represent all of the spatio-temporal variation in the snowmelt process across the catchment, and that some driving processes (i.e. radiation induced events) are also not represented adequately in either snow model. Either improved empirical relationships or a greater physical component to snowmelt modelling would be beneficial in this respect for future research. Accounting for snow redistribution by avalanches or wind as described in Freudiger et al. (2017) should also be investigated as this would potentially reduce some of the relatively large accumulations of snow that were found to persist over multiple years on some cells in the upper catchment during the reference period. Inaccuracies in the meteorological fields (METEO$_{OBS}$) that were used for the bias correction could also have caused some of the seasonal over- and underestimations in the hydrological regime. As discussed in Jobst et al. (2017) the climate network in the upper Clutha is sparse with very few sites located in medium to high elevations. Notwithstanding the improved representation of temperature provided by the Jobst et al. (2017) dataset compared to other products, the remaining biases in this temperature field would have also propagated into the bias corrected RCM fields and the corresponding hydrological baseline simulations.

For the two future periods (2050s and 2090s) the projections revealed substantial increases in runoff from May to October, and a decline between November and March. The dominant drivers behind this regime shift were changes in the seasonal distribution of precipitation (for the 2090s-winter +29 to +84%) and a rise in air temperature causing decreases in the seasonal

snow storage. These findings are mostly consistent with previous New Zealand based climate change impact assessments. In a New Zealand wide study Hendrikx et al. (2012) also modelled substantial reductions in the peak snow accumulations along the Southern Alps, which they attributed to decreases in the fraction of solid precipitation due to increases in air temperature. Using a semi-distributed hydrological model (TopNet) and ensemble of 12 CMIP-3 GCMs (including the four used herein),

Poyck et al. (2011) and Gawith et al. (2012) found a similar ensemble-mean increase in winter streamflow in the 2090s (for Balclutha and the upper Clutha – Matukituki catchment), although only relatively small decreases in summer river flow. In the upper Waitaki catchment (9490 km$^2$, located north-east of the Clutha and also with headwaters bordering the Main Divide of the Southern Alps), Caruso et al. (2017) found comparable large increases in lake inflows during winter (i.e. 76% for August) and a noticeable decease in summer (i.e. -13% for February) using the same hydrological model and GCM ensemble as Poyck

et al. (2011). An increase in winter precipitation was also identified as the main driver for the Waitaki.

Globally, similar changes in streamflow have been reported for many alpine catchments, for example in British Columbia (Mandal and Simonovic, 2017), Oregon (Chang and Jung, 2010) and the Austrian Alps (Laghari et al., 2012). In addition to increased winter precipitation, a reduction in solid precipitation is often reported to lead to an earlier melt peak and further enhanced winter flow (Kundzewicz, 2008). Here, a decrease in the proportion of solid precipitation combined with an

intensification of snowmelt was also found to contribute to more flow being generated during winter and spring, but the main driver remained the increase in winter precipitation.

Analysis of the uncertainty in the hydrological projections for the upper Clutha (Figure 12 and 13) showed that although the total spread of hydrological projections was large (i.e. increase of 40-116% for winter), for most seasons (except autumn) the direction of change was found to be consistent amongst individual members (increases in winter and spring, decreases in

summer). The main contributors to the spread in the projections for seasonal flow were (in ascending order): snow model (3-10%), bias correction method (4-22%), emission scenario (16-49%) and GCM (44-57%). As in this study, a large body of existing hydrological impact studies also identified GCM structure as the dominant source of uncertainty (e.g. Kingston and Taylor, 2010; Hughes et al., 2011; Teng et al., 2012; Thompson et al., 2014). It should be noted that the four GCMs constitute a subset of a total of 12 GCMs which had been previously selected by NIWA on the basis of a performance assessment for the

South-Pacific region (Ministry for the Environment, 2008). In terms of Tmean signal (A1B) the four GCMs had the 5[th], 10[th], 11[th] and 12[th] highest warming– hence the A1B projections used in this study are at the lower end of the "full" GCM envelope. A large part of the GCM related uncertainty was found to be caused by the precipitation signal, which became especially uncertain during the winter season. This finding is in agreement with a number of studies targeting alpine catchments such as the Hindu-Kush-Himalayan region (Palazzi et al., 2014; Lutz et al., 2016), the Pacific Northwest of the US (Jung et al., 2012),

Western Oregon (Chang and Jung, 2010) and the Southern Alps of New Zealand (Zammit and Woods, 2011). Hence constraining and accounting for the uncertainty associated with the precipitation output of GCMs and RCMs remains a major research challenge in hydrological impact studies.

Emission uncertainty was identified as the second most important source during most seasons, while in winter bias correction was found to introduce a similar level of uncertainty. These findings generally agree with the study of Prudhomme and Davies

(2009), in which emission scenario and downscaling (RCM vs. statistical method) uncertainty were of a comparable magnitude, but still considerably smaller when compared to GCM uncertainty. For alpine catchments in British Columbia the ranking order of uncertainty sources computed by Bennett et al. (2012) was also led by the GCM, followed by the emission scenario and in third hydrological parameter uncertainty.

As described in Kay and Davies (2008) and Thompson et al. (2014), different versions of the same hydrological model can be developed that differ in one particular routine (i.e. PET) allowing for a process specific uncertainty analysis. In the upper Clutha catchment, the high precipitation intensity in the headwaters combined with the relatively high proportion of snowmelt (~20%) means that the seasonal regime is largely controlled by the process of snowmelt rather than PET, which made the upper Clutha an appropriate candidate for the snow model specific uncertainty analysis. By using the two WaSiM versions

that only differ in their snowmelt routine the contribution of that process to the overall uncertainty could be assessed in isolation.

As expected, the contribution of the snow model was highest for winter (10%). However interestingly, the contribution of the snow model was still relatively high during summer (8%), a time of year when the influence of melt processes on streamflow were expected to be minor. This can likely be explained by the larger SWE volume that was modelled by Tindex (compared

to Anderson) during the summer reference period (Table 3). Thus, the Tindex SWE storage had the potential to release more melt water (compared to baseline) under the projected warming, which translated into an attenuation of the decrease in summer runoff. This is supported by the fact that the negative changes in summer runoff are consistently less pronounced for the Tindex model (not shown here). For autumn and spring, the snow model only added a small proportion (5 and 3%, respectively) to the overall uncertainty. Considering that spring is (historically) the main melt period, projections were expected to vary more

depending on the choice of the snow model. Hence for the spring season the results suggest that under the projected warming the effect of the snow model can be considered negligible especially when compared to the GCM and the emission scenario. At 10% of overall uncertainty in winter, the effect of the snow model is noticeable but substantially smaller than the variation caused by the GCM output (48%) (uncertainty of bias correction and emission scenario corresponds to 22 and 16%, respectively).

The study of Troin et al. (2016), which focused on the direct output of the snow model (i.e. SWE or duration of snow pack), came to comparable conclusions in the sense that hydrological models are not the major source of uncertainty for SWE projections. In their study, natural variability had a far greater effect on the projections for the individual snow indicators as the snow model component, which was shown here in a similar way for GCM structure.

## 5. Conclusions

The implementation of WaSiM for the Clutha River constitutes the first application of a fully distributed and grid-based hydrological model for climate change impact assessment in a large scale New Zealand catchment. The model chain that was built here to force WaSiM with RCM simulations can be regarded as an important contribution to the existing body of climate

change impact studies targeting snowmelt affected mid-latitude alpine catchments. The projections for the end of the 21$^{st}$ century encompass substantial increases in runoff for winter (71%) and spring (35%), while summer runoff is projected to decrease (-24%). The key drivers behind the changes in the regime were found to be an increase in winter precipitation and a reduction in the SWE storage between winter and spring. The changes in the regime will likely impact the capacity of the two

hydropower schemes (Clyde and Roxburgh; Figure 1) located downstream of Chards Rd, where production can be expected to increase from winter to early spring and decrease during the summer months.

Adopting the approach of Muerth et al. (2012), this study allowed the contribution of the individual uncertainty sources to be quantified in a more objective way opposed to a mere visual interpretation of results. For the first time the role of the rarely investigated snowmelt routine was explored together with three of the key uncertainty sources in hydrological impact studies

(i.e. GCM, emission scenario and bias correction method). While all components contributed to the total ensemble uncertainty, the selection of the GCM introduced the biggest spread to the range of runoff projections during all seasons. When looking at the climate signals (Figure 6 and 7) it becomes obvious that the uncertainty stemming from the precipitation signal (especially during winter) is the primary driver behind the large uncertainty in the hydrological projections and should therefore be the focus of future studies. In this context combining the limited number of RCM simulations with sophisticated statistical

techniques (e.g. the use of probability density functions as described in Tait et al. (2016)) could help to more fully explore the uncertainty range associated with the precipitation signal. Further, since the completion of this study additional RCM simulations based on RCP (Representative Concentration Pathways) scenarios and CMIP-5 GCMs have been generated for the New Zealand domain (Ministry for the Environment, 2016), which could be used to enlarge the existing ensemble of hydro-climatic projections for the Clutha.

The uncertainty linked to the snow model, which showed a pronounced seasonal variation (ranging from 3% in spring to 10% in winter), was found to be smaller when compared to the other components, but the findings of this study suggest that it should not be ignored as its effect was shown to be significant for both winter and summer runoff. Another important finding from this study is that the contribution of the snow model and the other model components to the overall uncertainty possesses a high inter-annual variability. While there was consistency regarding the main uncertainty source (i.e. GCM structure), the

second largest contributor varied between emission scenario and bias correction for the individual seasons. Future work should investigate if the selection of the snow model has a stronger impact in other regions or catchments of different size (i.e. small headwater sub-catchments). Model parameter uncertainty was not accounted for in this study but should be investigated as part of future work, which could help to understand and potentially improve misrepresentations in the historic runoff regime. The use of other hydrological indicators (i.e. low and high flow) should also be explored as the effect of the individual

components of the model chain might differ for such alternative metrics.

## Acknowledgements

We would like to thank a number of authorities who provided the various data sets used in this study. The National Institute of Water and Atmospheric Research (NIWA) for climate observations and the RCM simulations. Streamflow records were kindly provided by NIWA, Contact Energy Ltd. and the Otago Regional Council. The digital elevation model, soil data and land use information were obtained from Landcare Research New Zealand. This study was funded by a University of Otago Doctoral Scholarship and Publishing Bursary.

## Competing interests

The authors declare that they have no conflict of interest.

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

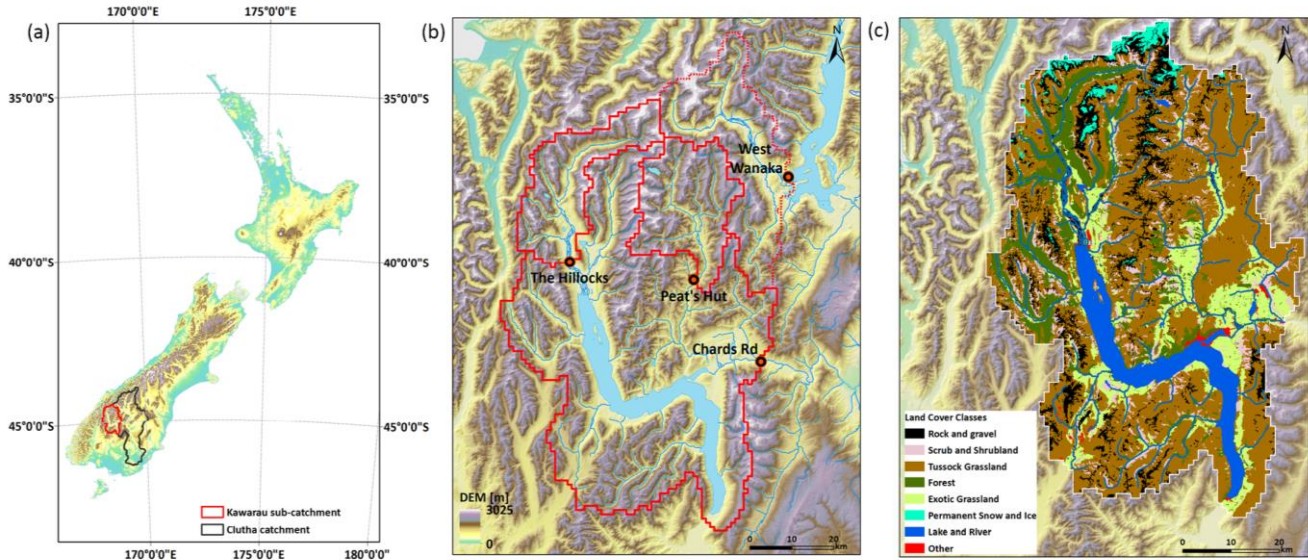

**Figure 1 Maps showing (a) New Zealand with the Clutha catchment located in the lower South Island, (b) the sub-catchments of the gauges recording discharge used in this study (note that the West Wanaka sub-catchment is outside the Kawarau catchment) and (c) a land cover classification of the Kawarau sub-catchment based on New Zealand's Land Cover Database (LCDB v3.0) that was published by Landcare Research in 2012.**

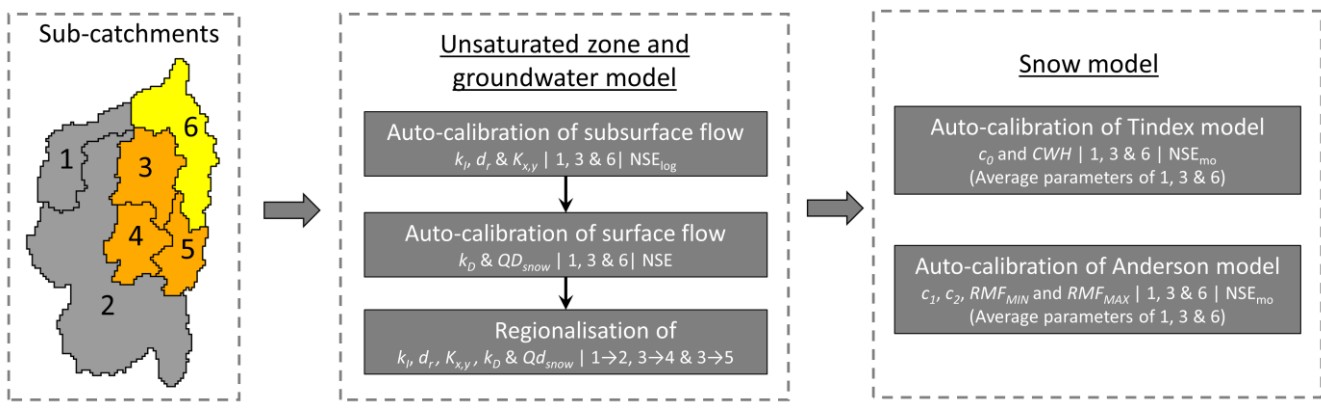

**Figure 2 The calibration workflow that was used to calibrate WaSiM. For each calibration step (either manual or automatic) information is provided as follows: "calibrated parameters | calibrated sub-catchments | performance criterium used". The parameters controlling subsurface flow encompass the drainage density for interflow ($dr$), the storage coefficient of interflow ($k_I$) and the groundwater conductivity in x or y-direction. Surface flow is controlled by the storage coefficient of surface runoff ($k_D$) and the fraction of snowmelt directly becoming surface runoff ($QD_{snow}$). The parameters of the snow models are shown in Table 1.**

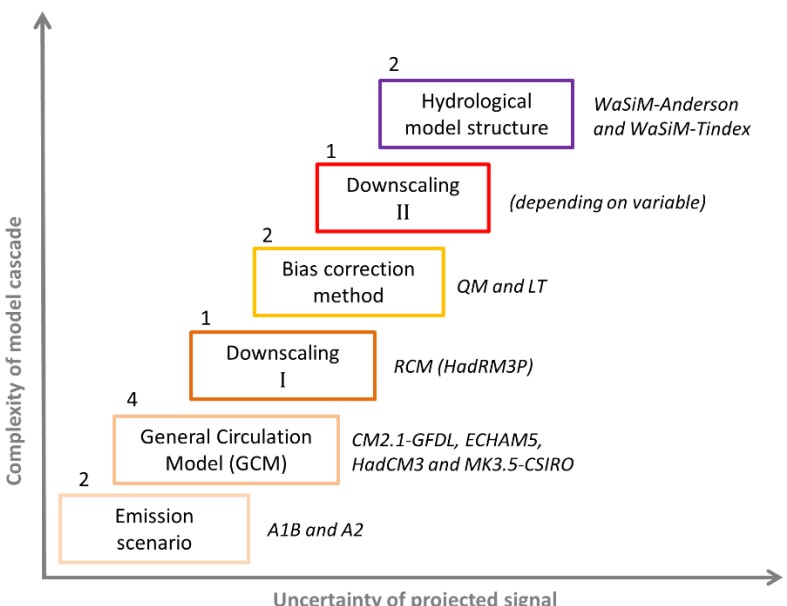

**Figure 3 The model cascade is depicted with the individual members of each component listed on the right. Permuting the members of all components results in a total of 32 hydrological projections.**

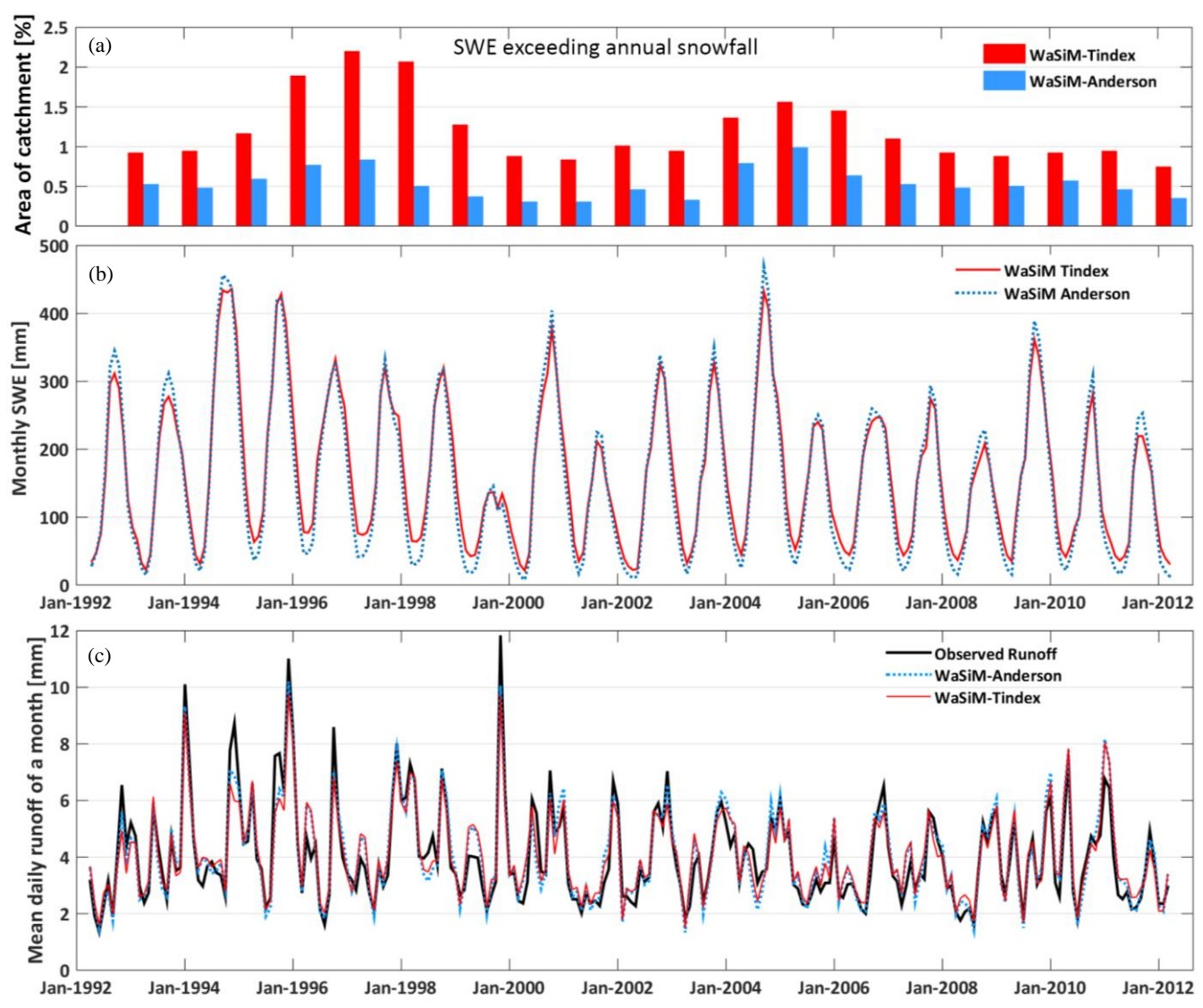

**Figure 4 (a) relative area of the Kawarau catchment where SWE exceeds annual snowfall for a hydrological year (e.g. 1.4.1992 – 31.3.1993) as an indicator for unrealistic build-up of snow (i.e. snow towers). (b) Modelled mean monthly snow water equivalent (SWE) based on the Tindex and Anderson simulations. (c) Observed and modelled (WaSiM-Anderson and WaSiM-Tindex) monthly runoff at Chards Rd for the reference period (1992-2012).**

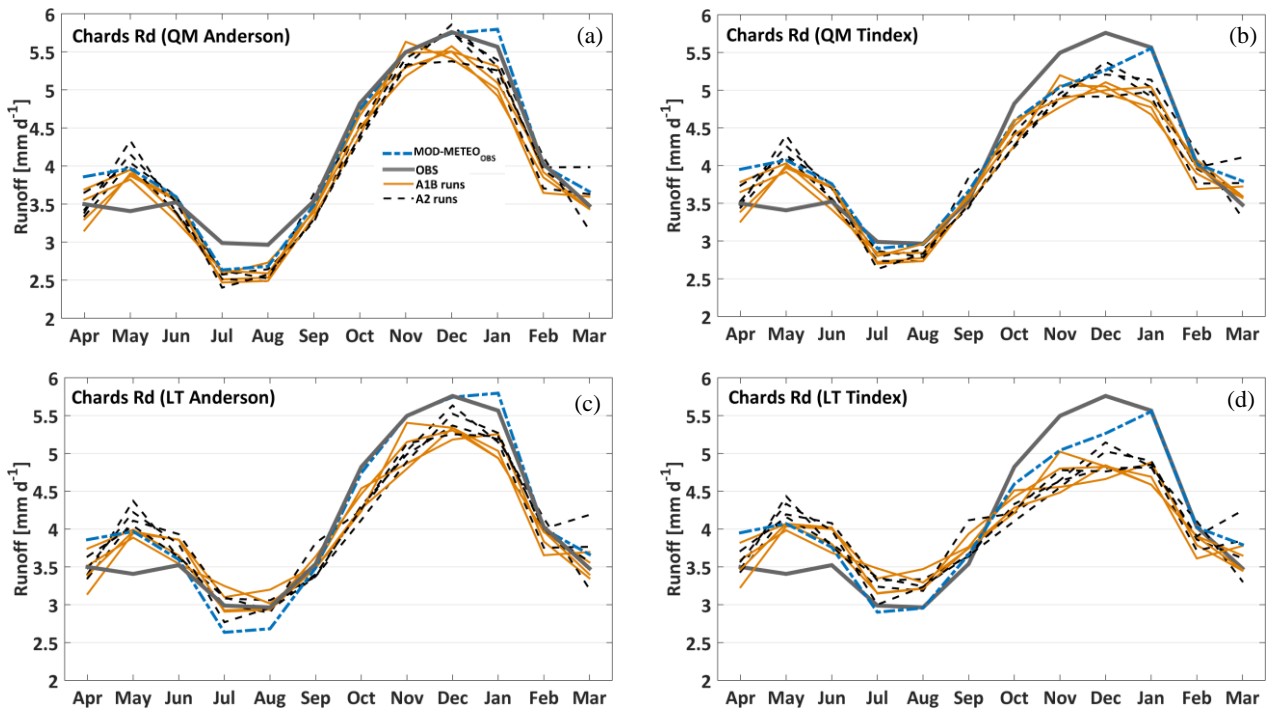

**Figure 5 RCM driven mean daily runoff simulations for the 1/4/1992–30/3/2011 period at Chards Rd. Simulations are compared with the observed regime (grey line) and the modelled (WaSiM forced with observed meteorology) regime (blue line) for the four subensembles: QM-Anderson (a), QM-Tindex (b), LT-Anderson (c) and LT-Tindex (d).**

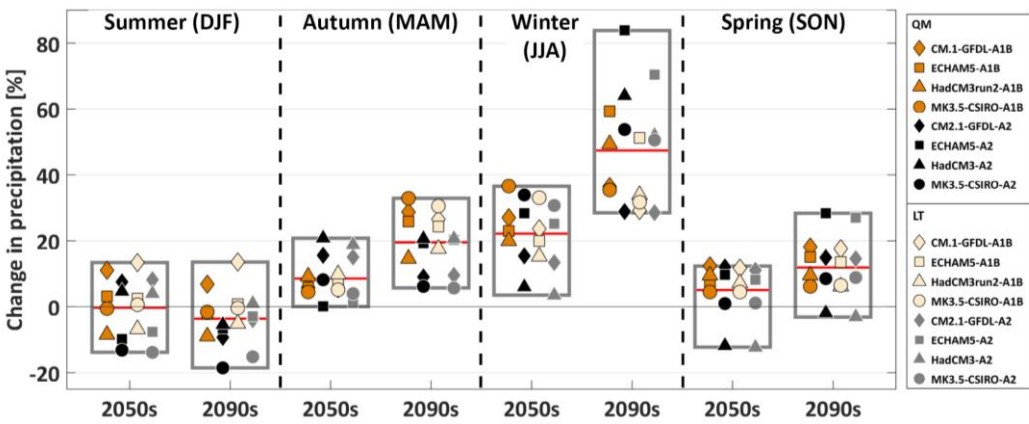

**Figure 6 The uncertainty range of the precipitation signal (domain average) is shown for the entire ensemble and the four seasons.For each box the uncertainty range is broken down into the two emission scenarios and the two bias correction methods. The red line represents the median of all 16 members.**

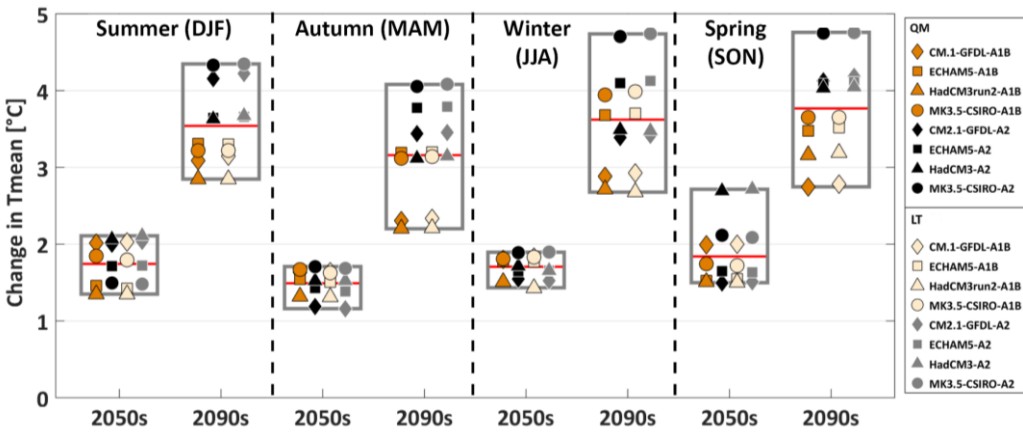

**Figure 7 The uncertainty range of the Tmean signal is shown for the entire ensemble and the four seasons. For each box the uncertainty range is broken down into the two emission scenarios and the two bias correction methods. The red line represents the median of all 16 members within a box.**

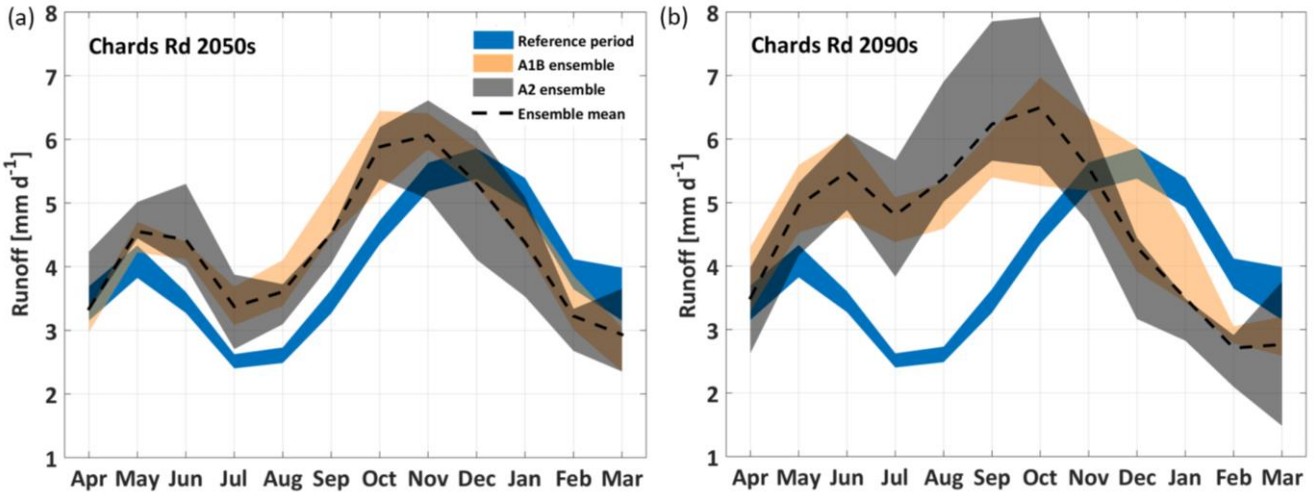

**Figure 8 Mean daily modelled runoff (orange envelope = A1B and grey envelope = A2) at Chards Rd during the 2050s (a) and 2090s (b) is compared with the historic simulations (blue). All simulations are based on the QM-Anderson subensemble.**

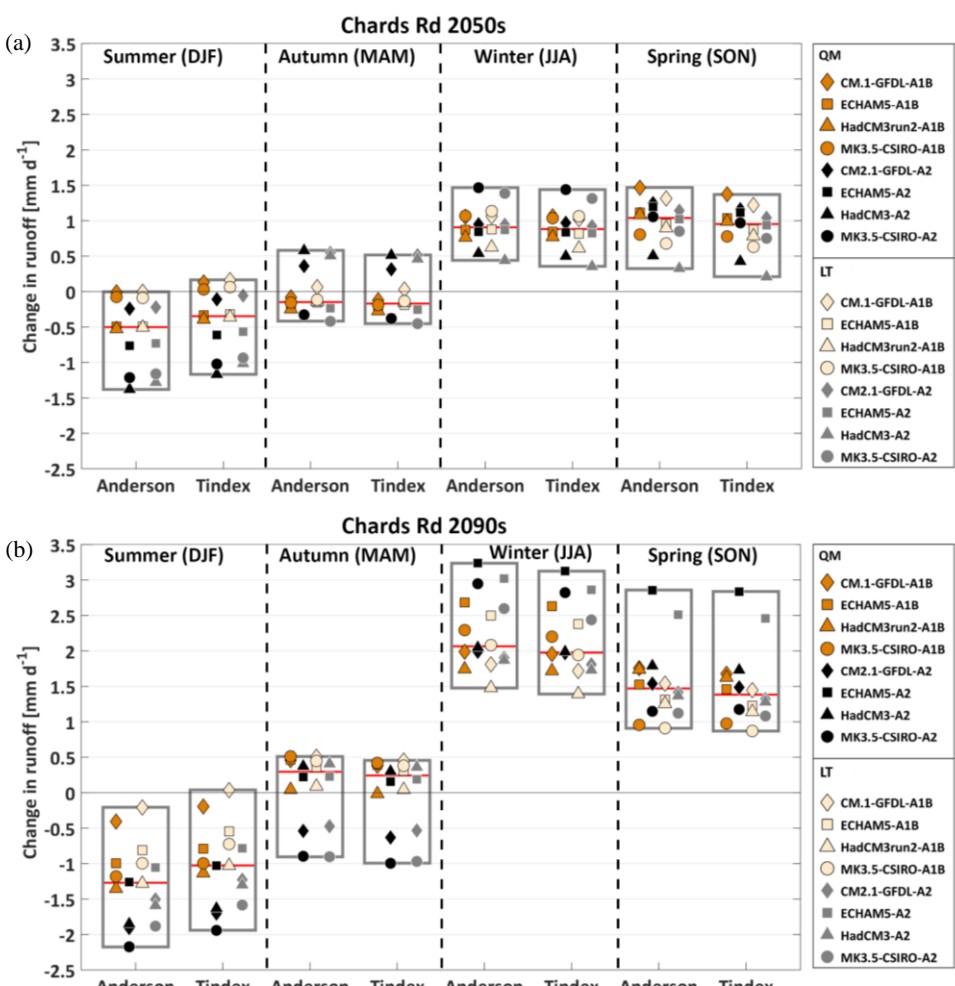

**Figure 9** The uncertainty range of the seasonal runoff signal at Chards Rd is shown for (a) the 2050s and (b) the 2090s. For each season an uncertainty box is shown for WaSiM-Anderson and WaSiM-Tindex simulations, respectively. In each box (red line = median) the uncertainty range is broken down into the two emission scenarios and the two bias correction methods (LT and QM).

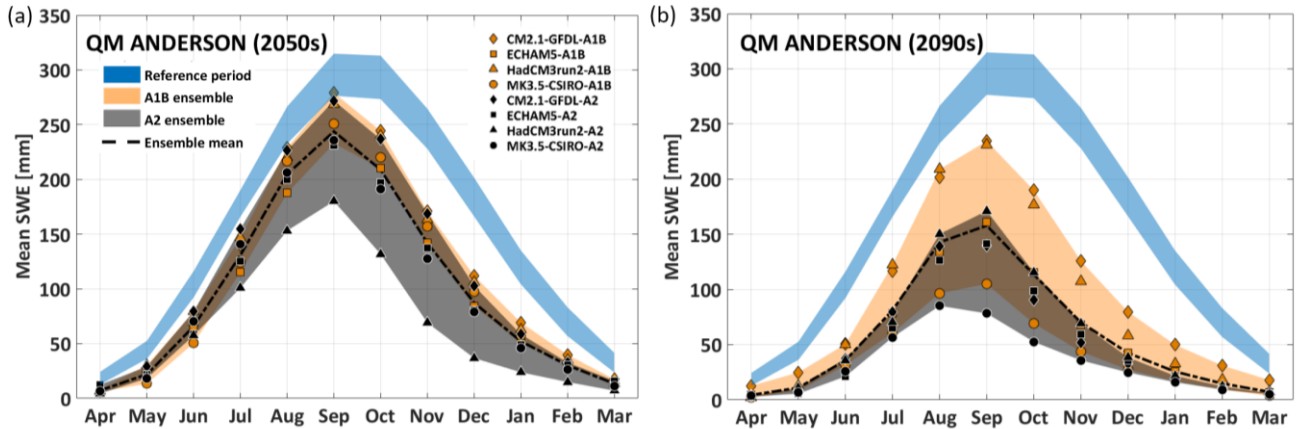

**Figure 10** The monthly SWE volume simulated by the QM-Anderson subensemble (km³) is depicted for (a) the 2050s and (b) the 2090s period. The individual members are augmented by the A1B envelope, the A2 envelope and the ensemble mean (dashed line). The blue envelope represents the range of SWE volume simulations during the baseline period.

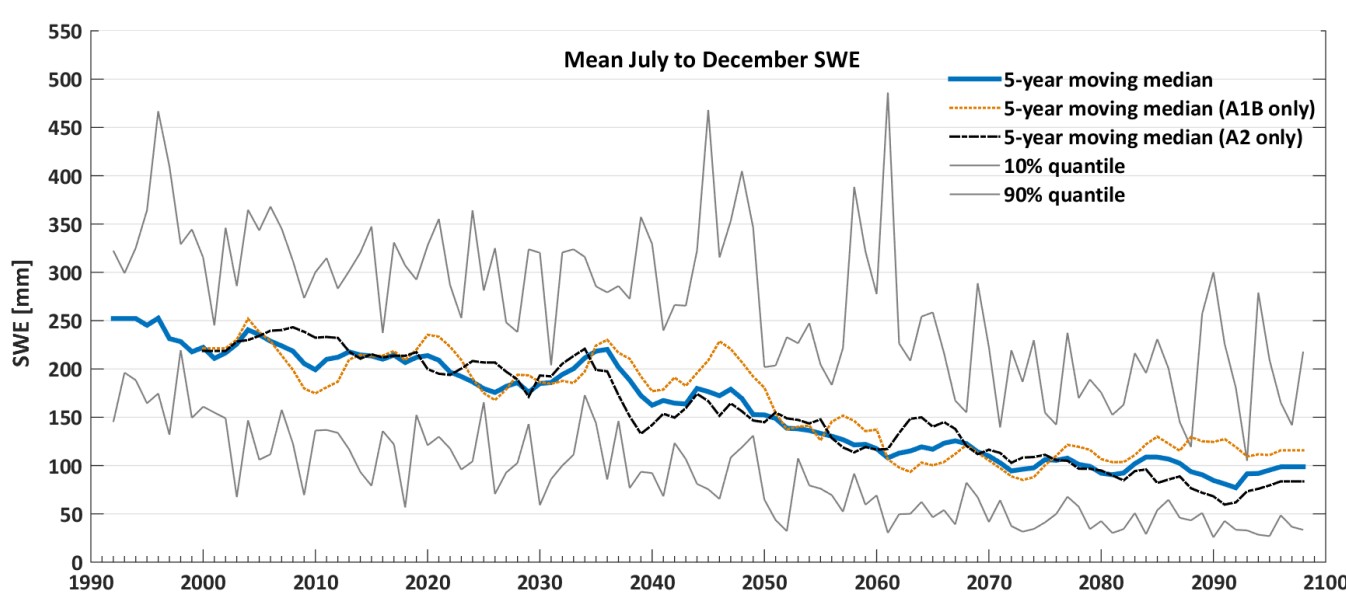

**Figure 11** The mean SWE of the six months with the historically highest SWE (July to December) is shown for the 16 transient QM simulations (including QM-Anderson and QM-Tindex).

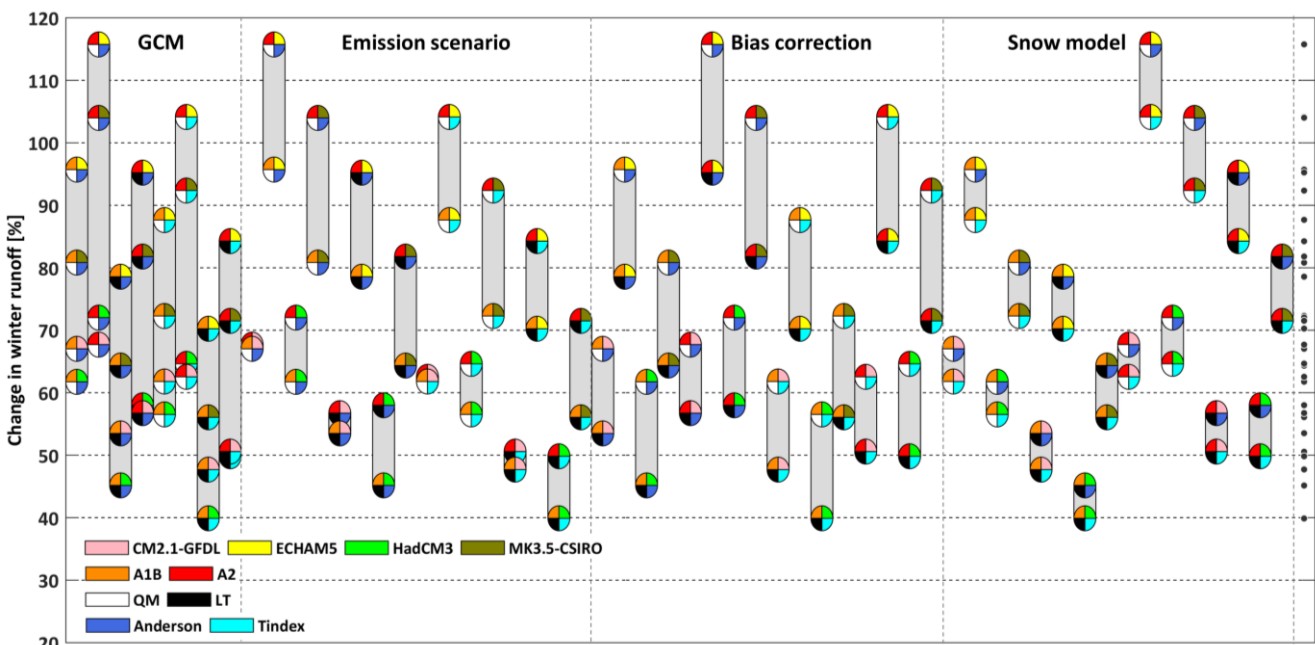

**Figure 12** Uncertainty contributions are isolated for each of the four model components GCM, emission scenario, bias correction method and snow model (the approach is described in more detail in section 3.4). The number of points in a bar corresponds to the number of members of the selected model component (e.g. four GCMs), while the number of bars is equal to the possible permutations based on the remaining three model components (2•2•2=8).

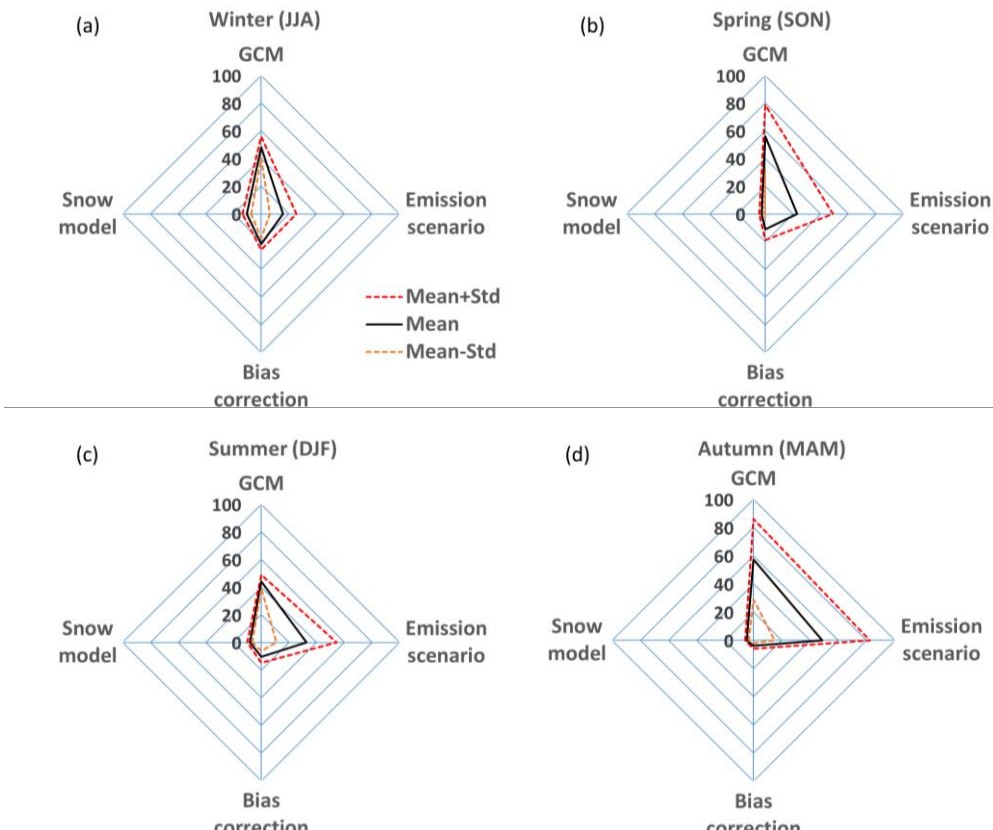

**Figure 13 Relative contribution of the four uncertainty sources to the overall uncertainty of the seasonal runoff signal for (b) winter, (c) spring, (d) summer and (e) autumn. Based on the permutations (as exemplified for winter in Figure 10) the radar charts show the mean contributions [%] (mean length of bars shown in Figure 10) of the four model components to the overall uncertainty as well as the standard deviations [%] (standard deviation of bars shown in Figure 10).**

**Table 1 Parameters of the two snowmelt routines as determined by the calibration routine. $T_0$ and CWH are mutual parameters of both models. The snow accumulation parameters are based on the findings of Auer (1974).**

| Anderson and Tindex | | |
|---|---|---|
| Threshold temperature for snowmelt ($T_0$) | 0.00 | [°C] |
| Water holding capacity of snow (*CWH*) | 0.29 | [-] |
| **Tindex** | | |
| Temperature dependent DDF ($c_0$) | 1.91 | [mm °C$^{-1}$ d$^{-1}$] |
| **Anderson** | | |
| Temperature dependent DDF ($c_1$) | 0.63 | [mm °C$^{-1}$ d$^{-1}$] |
| Wind dependent DDF ($c_2$) | 0.08 | [mm (°C m/s d)$^{-1}$] |
| Minimum radiation melt factor ($RMF_{MAX}$) | 3.13 | [mm °C$^{-1}$ d$^{-1}$] |
| Maximum radiation melt factor ($RMF_{MIN}$) | 0.36 | [mm °C$^{-1}$ d$^{-1}$] |
| **Snow accumulation** | | |
| Temperature, at which 50% of precipitation are falling as snow ($T_{R/S}$) | 3.00 | [°C] |
| Half of the temperature-transition range from snow to rain ($T_{trans}$) | 3.00 | [K] |

**Table 2 Nash Sutcliffe values based on daily (NSE), logarithmic (NSE$_{log}$) and monthly streamflow data during the calibration (Cal) and validation (Val) periods calculated for the WaSiM-Anderson and WaSiM-Tindex (in brackets) simulations. Note the different validation periods for The Hillocks and Peat's Hut due to shorter records.**

| River | Gauge | Cal (1.4.2008-31.3.2012) | | | Val (1.4.1992-31.3.2008) | | |
|---|---|---|---|---|---|---|---|
| | | NSE | NSE$_{log}$ | NSE$_{mo}$ | NSE | NSE$_{log}$ | NSE$_{mo}$ |
| Dart | The Hillocks (1996-2012) | 0.77 (0.77) | 0.77 (0.78) | 0.92 (0.92) | 0.64 (0.65) | 0.64 (0.68) | 0.78 (0.79) |
| Shotover | Peat's Hut (1996-2012) | 0.64 (0.65) | 0.67 (0.70) | 0.81 (0.82) | 0.60 (0.62) | 0.65 (0.70) | 0.76 (0.79) |
| Kawarau | Chards Rd | 0.87 (0.88) | 0.88 (0.87) | 0.89 (0.90) | 0.87 (0.85) | 0.86 (0.86) | 0.89 (0.87) |
| Matukituki | West Wanaka | 0.67 (0.67) | 0.64 (0.65) | 0.80 (0.80) | 0.62 (0.62) | 0.72 (0.72) | 0.83 (0.82) |

**Table 3 The historic (1/4/1992 – 30/3/2011) water balance terms of the MOD-METEO$_{OBS}$ run compared with the corresponding (depending on snow model) RCM ensemble means (QM and LT). The seasonal and annual SWE volumes are also shown.**

| Anderson | P | ET | Q | SWE [mm] | | | | |
|---|---|---|---|---|---|---|---|---|
| | [mm] | [mm] | [mm] | DJF | MAM | JJA | SON | YEAR |
| MOD-METEO$_{OBS}$ | 2007 | 471 | 1512 | 117 | 33 | 183 | 286 | 154 |
| QM | 1926 | 473 | 1425 | 123 | 31 | 176 | 277 | 152 |
| LT | 1931 | 466 | 1443 | 117 | 33 | 150 | 240 | 134 |
| **Tindex** | | | | | | | | |
| MOD-METEO$_{OBS}$ | 2007 | 473 | 1509 | 143 | 53 | 181 | 275 | 163 |
| QM | 1926 | 475 | 1422 | 143 | 44 | 172 | 262 | 154 |
| LT | 1931 | 468 | 1441 | 126 | 42 | 139 | 216 | 130 |