# Peer review of "Intercomparison of different uncertainty sources in hydrological climate change projections for an alpine catchment (upper Clutha River, New Zealand)"

_Hydrology and Earth System Sciences, 2017_

## Referee Comment (RC1) · Anonymous Referee #1 · 29 Nov 2017

> General comments:

The study evaluates different sources of uncertainty in hydrological projections with focus on the impact of the snow model. The experiments were carried out for a sub-catchment of the Clutha River, New Zealand. It is nicely demonstrated that the snow model contributes to the uncertainty of seasonal variations and high inter-annual variability. Despite that the choices of the global climate model, emission scenario, and bias correction method generally have more impact on the total uncertainty, the results show that the uncertainty linked to the snow model should not be neglected in alpine

catchments. It is a compact study with a good structure. The manuscript is clearly written, and I enjoyed reading it. Some minor comments might be considered to clarify a few parts. Including:

> Specific comments:

(1) Study area:

The studied area is called the "Clutha catchment" and is introduced as a "major" and "large-scale New Zealand catchment". But – according to Chapter 2.1 – the analysis was carried out only for the most north-western sub-catchment (outlet Chards Rd). To clarify the whole study, the authors should please rephrase those parts of the manuscript (see also technical corrections) which lead to the misunderstanding that the results represent the whole Clutha catchment.

(2) Catchment characteristics:

Please provide some information about i.e. elevation range, glacier extent, vegetation of the sub-catchment (outlet Chards Rd).

(3) Snow models:

The study focusses on snowmelt uncertainty for hydrological projections and should therefore give more details about the two types of snow models used here. For example, it would be helpful to know if all snow is gone after the summer (except for glaciers) or if snow accumulates from year to year.

(4) Baseline:

Please introduce the baseline model in "Data and methods". How has it been calibrated?

(5) Results – Baseline simulations:

Apart from the graphs, values of model efficiency would strengthen the results.

(6) The climate change signals:

Please clarify in the text (and legends) "temperature signal" / "change in Tmean".

(7) Streamflow signal:

Some parts could better be included in the discussion.

> Technical corrections:

(8) P.3, l. 4 : makes it

(9) P.5, l. 13 : runoff regime

(10) P.6, l. 11 : Clutha catchment or north-western sub-catchment?

(11) P.6, l. 30 : most striking transformation for the 2090s

(12) P.7, l. 18 : Clutha catchment or north-western sub-catchment?

(13) P.8, l. 28 – 32 : see (7)

(14) Table 1 : provide units to the reader

(15) Figure 1b : increase the size of the coordinates and the catchment border

(16) Figure 1c : show the sub-catchments used for the calibration of the snow model

(17) Figure 3 : rescaling of the y-axis (2 to 6mm/d) and different colors for the single lines would help the reader

(18) Figure 5 : see point (6)

(19) Figure 8 : provide a legend for the different shapes

(20) Figure 9 : Please increase the size of the circles and/or change the colors since it is really hard to distinguish between the colors.
* * *
598, 2017.

---

## Referee Comment (RC2) · H. McMillan (Referee) · 7 Jan 2018

Review of "Intercomparison of different uncertainty sources in hydrological climate change projections for an alpine catchment (Clutha River, New Zealand)" by Jobst et al.

This paper is an exploration of the potential impacts of climate change on precipitation, snowpack and river flow in a sub-catchment of the Clutha River in New Zealand. The paper is well written and comprehensive, and I recommend it for publication after the

minor revisions outlined below.

1. The start of the paper makes much of the large size and representativeness of the Clutha catchment. Given that the study is actually only carried out on a sub catchment (less than one quarter area) of the Clutha, which does not include any of the drier Otago climate described, I suggest this section be revised for relevance.

2. p5 L15 The authors should define in the text what they mean by the "observed regime" (i.e. monthly flow values averaged across all years) so that the readers are clear what is being evaluated. Similarly the meaning of "summer climate change signal" should be defined.

3. Section 2.2. The largest comment that I have on the paper is that there is insufficient information/discussion to convince the reader that the WaSim hydrology model does a good job of representing the catchment. Trust in this model is essential for the uncertainty analysis and conclusions of the paper. There is a brief mention of Nash Sutcliffe values at p4 L20, but the addition of a hydrograph plot showing modelled/observed values for some suitable period would make this more convincing. Especially given that Fig 3 shows significant under prediction of winter flow, and it is unclear what causes this problem. The range of simulations in the paper do not envelope the observed flow – so there is some uncertainty that is unaccounted for in the paper and I am left wondering where it is? Some additional discussion is warranted here, including discussion of potential uncertainty in hydrologic model parameters.

4. Section 2.3. Worth noting that the climate scenarios used for New Zealand have now been superseded by 6 RCM*4 RCP Scenarios with CMIP-5 GCMs and a new bias correction that improves on quantile correction. See: http://ccii.org.nz/wp-content/uploads/2016/10/RA1-Synthesis-report.pdf For the next paper perhaps!

5. Fig 6. I don't understand the comment about different y-axes.

6. Fig 9. Please include an explanation of what a radar chart shows.

7. p9 It would be useful to reference this paper:

Hendrikx, J., Hreinsson, E.Ö., Clark, M.P. and Mullan, A.B., 2012. The potential impact of climate change on seasonal snow in New Zealand: part I—an analysis using 12 GCMs. Theoretical and Applied Climatology, 110(4), pp.607-618.

———————————————

---

## Author Comment (AC1) · 1 Feb 2018

General comments: The study evaluates different sources of uncertainty in hydrological projections with focus on the impact of the snow model. The experiments were carried out for a subcatchment of the Clutha River, New Zealand. It is nicely demonstrated that the snow model contributes to the uncertainty of seasonal variations and high inter-annual variability. Despite that the choices of the global climate model, emission scenario, and bias correction method generally have more impact on the total uncertainty, the results show that the uncertainty linked to the snow model should not be neglected in alpine catchments. It is a compact study with a good structure. The manuscript is clearly written, and I enjoyed reading it. Some minor comments might be considered to clarify a few parts. Including:

(1) Study area: The studied area is called the "Clutha catchment" and is introduced as a "major" and "large-scale New Zealand catchment". But – according to Chapter 2.1 – the analysis was carried out only for the most north-western sub-catchment (outlet Chards Rd). To clarify the whole study, the authors should please rephrase those parts of the manuscript (see also technical corrections) which lead to the misunderstanding that the results represent the whole Clutha catchment.

**Response:** *In the paper we now state more clearly that a hydrological model was developed for the entire Clutha (as described in Jobst 2017) and that this paper focuses on the unmanaged north-western part of the Clutha. Outflow of the Kawarau sub-catchment (focus of this study) is highly correlated with the other unmanaged headwater tributaries and as most of the central Clutha catchment is characterised by a dry climate (with comparably little additional streamflow generated) the results for the Kawarau can be considered broadly representative of the main stem of the Clutha (including the outlet Balclutha, Figure 1).*

*However we agree that the main focus of this study is on the Kawarau sub-catchment and therefore "Clutha catchment" has been replaced with north-western sub-catchment or similar in most parts of the document. Corresponding changes were made to the following lines in the paper:*

*-P1 L12*
*-P3 L5, L10, L11, L23-24, L27-28 and L30-32*
*-P6 L22*
*-P8 L16*
*-P11 L12*

*Further, Figures 5,6 and 9 (note old figure number plus 1) were updated as they showed data for the entire Clutha catchment. They now show averaged data for the sub-catchment (Chards Rd) only. The relative signals shown in the updated figures are very similar to the previous ones for the entire Clutha and therefore only minor edits had to be added to the results sections of this document.*

(2) Catchment characteristics: Please provide some information about i.e. elevation range, glacier extent, vegetation of the sub-catchment (outlet Chards Rd).

*Response: The requested statistics and catchment characteristics have been added to the document (P3 L32 – P4 L4)*

*"Most of the Kawarau sub-catchment is covered by indigenous tussock grassland followed by low producing exotic grassland and indigenous forest. The elevation of the sub-catchment ranges between 300 and 2800 m with an ice cover of approximately 84 km² which corresponds to 55% of the Clutha catchment's ice cover (New Zealand's Land Cover Database v3.0 as published by Landcare Research in 2012)."*

(3) Snow models: The study focusses on snowmelt uncertainty for hydrological projections and should therefore give more details about the two types of snow models used here. For example, it would be helpful to know if all snow is gone after the summer (except for glaciers) or if snow accumulates from year to year.

*Response: A new figure has been added that shows the seasonal variation of SWE and how some of the snow persists over summer and into autumn (Figure 3).*

[Figure]

*Some more details about the snow models have been added to 2.2 (P4 L15-19)*

*"The Tindex model calculates the melting rate via a degree-day factor multiplied by the difference between actual temperature and the melting point temperature. The Anderson model is more complex as it computes four separate melt fractions and accounts for radiation by using a seasonal melt factor. Further the Anderson approach also models the refreezing of liquid water stored in the snow pack if the actual temperature is below melting point."*

(4) Baseline: Please introduce the baseline model in "Data and methods". How has it been calibrated?

*Response: We decided not to add any more detail on this matter because of the following reason. The calibration of the hydro model has already been described as part of 2.2 (P4 L20-31). The baseline simulations presented in 3.1 were not calibrated per se but are the product of WaSiM being forced by the bias corrected and downscaled meteorological variables of the RCM simulations.*

(5) Results – Baseline simulations: Apart from the graphs, values of model efficiency would strengthen the results.

*Response: A new figure has been added (Figure 3) showing daily simulations of discharge forced by the observed meteorology during the validation period. A new table (Table 2) with more NSE values was also added to the document. Also see second Reviewer's comment 3.*

[Figure]

| River | Gauge | Cal (1.4.2008-31.3.2012) | | | Val (1.4.1992-31.3.2008) | | |
|---|---|---|---|---|---|---|---|
| | | NSE | $NSE_{log}$ | $NSE_{mo}$ | NSE | $NSE_{log}$ | $NSE_{mo}$ |
| Dart | The Hillocks (1996-2012) | 0.77 (0.77) | 0.77 (0.78) | 0.92 (0.92) | 0.64 (0.65) | 0.64 (0.68) | 0.78 (0.79) |
| Shotover | Peat's Hut (1996-2012) | 0.64 (0.65) | 0.67 (0.70) | 0.81 (0.82) | 0.60 (0.62) | 0.65 (0.70) | 0.76 (0.79) |
| Kawarau | Chards Rd | 0.87 (0.88) | 0.88 (0.87) | 0.89 (0.90) | 0.87 (0.85) | 0.86 (0.86) | 0.89 (0.87) |
| Matukituki | West Wanaka | 0.67 (0.67) | 0.64 (0.65) | 0.80 (0.80) | 0.62 (0.62) | 0.72 (0.72) | 0.83 (0.82) |

(6) The climate change signals: Please clarify in the text (and legends) "temperature signal" / "change in Tmean".

*Response: Mean air temperature has been abbreviated with Tmean throughout the document*

(7) Streamflow signal: Some parts could better be included in the discussion.

*Response: To avoid unnecessary repetitions and to keep this paper concise we would like to leave section 3.3.1 as it currently is. Adding individual sentences to the discussion would require reintroducing the figure and what the data points depict.*

> Technical corrections:

(8) P.3, l. 4 : makes it

*Changed as suggested*

(9) P.5, l. 13 : runoff regime

*Changed as suggested*

(10) P.6, l. 11 : Clutha catchment or north-western sub-catchment?

*Changed as suggested*

(11) P.6, l. 30 : most striking transformation for the 2090s

*Changed as suggested*

(12) P.7, l. 18 : Clutha catchment or north-western sub-catchment?

*Changed as suggested*

(13) P.8, l. 28 – 32 : see (7)

*See comment under (7)*

(14) Table 1 : provide units to the reader

*Units were added to the table as suggested*

(15) Figure 1b : increase the size of the coordinates and the catchment border

*Changed as suggested*

(16) Figure 1c : show the sub-catchments used for the calibration of the snow model

*Changed as suggested*

(17) Figure 3 : rescaling of the y-axis (2 to 6mm/d) and different colors for the single lines would help the reader

*Changed as suggested. As the focus is on the groupings (i.e. A1B and A2 emission scenario runs) the same colors are supposed to be used for each grouping, while different colors for the individual lines would make these figures even harder to read.*

(18) Figure 5 : see point (6)

*Caption changed to: "mean temperature (Tmean) signal"*

(19) Figure 8 : provide a legend for the different shapes

*The different shapes are shown in Figure 8a to make this more obvious "The legend corresponding to all eight subplots is shown in Figure 8a." has been added to the caption.*

(20) Figure 9 : Please increase the size of the circles and/or change the colors since it is really hard to distinguish between the colors.

*Changed as suggested and we have also adjusted the layout of the figure*

---

## Author Comment (AC2) · 1 Feb 2018

*We would like to thank the second reviewer for the constructive comments. As already explained under our response to reviewer one we have clarified the geographical domain of this modelling study throughout the document and believe that the geographical focus of this study is now more obvious. We have also provided more detail about the snow models used, and presented more results from the validation of the hydrological model including some more performance criteria. Please see below for our individual responses.*

H. McMillan (Referee) hmcmillan@sdsu.edu Review of "Intercomparison of different uncertainty sources in hydrological climate change projections for an alpine catchment (Clutha River, New Zealand)" by Jobst et al. This paper is an exploration of the potential impacts of climate change on precipitation, snowpack and river flow in a sub-catchment of the Clutha River in New Zealand. The paper is well written and comprehensive, and I recommend it for publication after the

minor revisions outlined below.

1. The start of the paper makes much of the large size and representativeness of the Clutha catchment. Given that the study is actually only carried out on a sub catchment (less than one quarter area) of the Clutha, which does not include any of the drier Otago climate described, I suggest this section be revised for relevance.

***Response:*** *See answers to comment 1 of reviewer 1.*

2. p5 L15 The authors should define in the text what they mean by the "observed regime" (i.e. monthly flow values averaged across all years) so that the readers are clear what is being evaluated. Similarly the meaning of "summer climate change signal" should be defined.

***Response:*** *Sentence has been changed to: "The regimes (i.e. monthly flow values averaged across all years) of the eight RCM driven simulations…"*

*The following sentence has also been added P7 L5-6 "The climate change signals of Tmean and precipitation that are presented in the following section correspond to the mean change of the spatio-temporal average between a future (either 2050s or 2090s) and the reference period."*

3. Section 2.2. The largest comment that I have on the paper is that there is insufficient information/discussion to convince the reader that the WaSim hydrology model does a good job of representing the catchment. Trust in this model is essential for the uncertainty analysis and conclusions of the paper. There is a brief mention of Nash Sutcliffe values at p4 L20, but the addition of a hydrograph plot showing modelled/observed values for some suitable period would make this more convincing. Especially given that Fig 3 shows significant under prediction of winter flow, and it is unclear what causes this problem.

***Response:*** *A new figure showing the hydrographs has been added (i.e. Figure 3) to 2.2 with an additional table that lists some more performance statistics for several hydro gauges. The description of the model performance is now also covered in more detail P4 L31 – P5 L15: "While daily NSE values were lower for the three tributaries (Dart, Shotover and Matukituki River), monthly NSE values indicated a good performance (Table 2). For the Matukituki River the validation of both WaSiM versions revealed a substantially better performance (monthly NSE of 0.83 and 0.82, respectively) when compared to the TopNet based modelling study of Gawith et al. (2012) (monthly NSE of 0.68). For Chards Rd the performance of both WaSiM-Anderson and WaSiM-Tindex revealed a strong performance at the daily and monthly time scale, with NSE values between 0.85 and 0.90 across all model versions, timescales and time periods (Table 2). The hydrographs of WaSiM-Anderson and*

*WaSiM-Tindex (Figure 3) further indicate a realistic representation of observed daily runoff at Chards Rd (only the first four years of the validation period are shown for clarity). Obvious inaccuracies of both WaSiM versions are an underestimation of larger flow events during the melt period (e.g. November-December 1994) and an overestimation during autumn (e.g. April-May 1994). The likeliest explanation is that not enough snow is being accumulated from autumn to early winter and consequently the main melt peaks are under-simulated. Inaccuracies are generally larger with WaSiM-Tindex as can be seen in December 1994 (Figure 3). Compared to WaSiM-Anderson the snow pack melts slower which causes a greater underestimation of runoff during the main melt period. Overall the visually better performance of WaSiM-Anderson for the 1992-1996 period is substantiated by the daily NSE values which correspond to 0.91 and 0.87 for WaSiM-Anderson and WaSiM-Tindex, respectively."*

[Figure]

| River | Gauge | Cal (1.4.2008-31.3.2012) | | | Val (1.4.1992-31.3.2008) | | |
|---|---|---|---|---|---|---|---|
| | | NSE | NSE$_{log}$ | NSE$_{mo}$ | NSE | NSE$_{log}$ | NSE$_{mo}$ |
| Dart | The Hillocks (1996-2012) | 0.77 (0.77) | 0.77 (0.78) | 0.92 (0.92) | 0.64 (0.65) | 0.64 (0.68) | 0.78 (0.79) |
| Shotover | Peat's Hut (1996-2012) | 0.64 (0.65) | 0.67 (0.70) | 0.81 (0.82) | 0.60 (0.62) | 0.65 (0.70) | 0.76 (0.79) |
| Kawarau | Chards Rd | 0.87 (0.88) | 0.88 (0.87) | 0.89 (0.90) | 0.87 (0.85) | 0.86 (0.86) | 0.89 (0.87) |
| Matukituki | West Wanaka | 0.67 (0.67) | 0.64 (0.65) | 0.80 (0.80) | 0.62 (0.62) | 0.72 (0.72) | 0.83 (0.82) |

The range of simulations in the paper do not envelope the observed flow – so there is some uncertainty that is unaccounted for in the paper and I am left wondering where it is? Some additional discussion is warranted here, including discussion of potential uncertainty in hydrologic model parameters.

***Response:*** *See above and the following sentence (P13 L23-25) "Model parameter uncertainty was not accounted for in this study but should be part of future work, which could help to understand and potentially improve misrepresentations in the historic streamflow regime."*

*P10 L20-31:"* Overall the baseline analysis showed that the individual sub-ensembles performed differently and that the observed regime was not always enveloped by the corresponding range of simulations. This introduces some additional uncertainty into the projections that could not be quantified or accounted for in this study. A potential explanation is that neither snow model was able to accurately represent all of the spatio-temporal variation in the snowmelt process across the catchment, and that some driving processes (i.e. radiation induced events) are also not represented adequately in either snow model. Either improved empirical relationships or a greater physical component to snowmelt modelling would be beneficial in this respect for future research.

Inaccuracies in the meteorological fields (METEO$_{OBS}$) that were used for the bias correction could also have caused some of the seasonal over- and underestimations in the hydrological regime. As discussed in Jobst et al. (2017) the climate network in the upper Clutha is sparse with very few sites located in medium to high elevations. Notwithstanding the improved representation of temperature provided by the Jobst et al. (2017) dataset compared to other products, the remaining biases in this temperature field would have also propagated into the bias corrected RCM fields and the corresponding hydrological baseline simulations.

4. Section 2.3. Worth noting that the climate scenarios used for New Zealand have now been superseded by 6 RCM*4 RCP Scenarios with CMIP-5 GCMs and a new bias correction that improves on quantile correction. See: http://ccii.org.nz/wpcontent/uploads/2016/10/RA1-Synthesis-report.pdf For the next paper perhaps!

*Response: We appreciate the suggestion of the author and have added the following comment to the conclusion section as part of future work P13 L12-15:" Since the completion of this study additional RCM simulations based on RCP (Representative Concentration Pathways) scenarios and CMIP-5 GCMs have been generated for the New Zealand domain (Ministry for the Environment, 2016), which could be used to enlarge the existing ensemble of hydro-climatic projections for the Clutha."*

5. Fig 6. I don't understand the comment about different y-axes.

*Response: Comment was unnecessary, agreed and deleted.*

6. Fig 9. Please include an explanation of what a radar chart shows.

*Response: Added: "Based on the permutations for each season the radar charts show the mean contributions [%] of the four model components to the overall uncertainty as well as the standard deviations [%]."*

7. p9 It would be useful to reference this paper: Hendrikx, J., Hreinsson, E.Ö., Clark, M.P. and Mullan, A.B., 2012. The potential impact of climate change on seasonal snow in New Zealand: part IâǍŤan analysis using 12 ˇ GCMs. Theoretical and Applied Climatology, 110(4), pp.607-618.

*Response: Agreed paper has been added to document P10 L29-31 : "In a New Zealand wide study Hendrikx et al. (2012) also modelled substantial reductions in the peak snow accumulations along the Southern Alps, which they attributed to decreases in the fraction of solid precipitation due to increases in air temperature."*

*General/additional response: We have also made some minor modifications throughout the document in order to avoid confusion related to the terminology of runoff/streamflow. As modelled streamflow (m3/s) was converted to runoff (mm) for better comparison with other studies we have consistently replaced the term streamflow with runoff in the main document when we are referring to the actual results (as presented in the corresponding figures).*

---

## Author Response (AR2)

Dear editor,

Thank you for your detailed comments. We have revised most of the figures (there are now less subplots to keep the overall figure count as low as possible), added some additional figures where requested and tried to make this paper more transparent. We added more background information on the calibration process that was used as part of the implementation of WaSiM and clarified the role of permanent snow and ice in the Kawarau catchment. The potential issue of snow towers accumulating along the main divide and the headwaters of the catchment was also investigated. We hope that this revised version of the paper now provides sufficient information on the hydrological model as well as the snow and ice stores during both the reference and the future periods.

Best regards,
Andreas Jobst (on behalf of all co-authors)

Editor Decision: Publish subject to revisions (further review by editor and referees) (10 Mar 2018) by Kerstin Stahl

Comments to the Author:

Dear authors,

thanks for the revisions. I found several of them satisfying, but when reading the paper in detail now, I found there are crucial remaining issues related to the reviewer comments, which I do not see addressed sufficiently in the replies and added material. Also there are number of smaller editorial aspects relating to figures etc. that need to be improved, before I can make a final decision. I will keep the option open to appoint another reviewer again to advise.

Referee 1 in points 2 *(original comment: "Catchment characteristics: Please provide some information about i.e. elevation range, glacier extent, vegetation of the sub-catchment (outlet Chards Rd)")* and 3 *(original comment: " Snow models: The study focusses on snowmelt uncertainty for hydrological projections and should therefore give more details about the two types of snow models used here. For example, it would be helpful to know if all snow is gone after the summer (except for glaciers) or if snow accumulates from year to year.")* raised questions about detailed land cover and about the potential influence of snow accumulation over the years. Referee 2 found that streamflow is not enveloped in the simulation *(original comment: "The range of simulations in the paper do not envelope the observed flow – so there is some uncertainty that is unaccounted for in the paper and I am left wondering where it is? Some additional discussion is warranted here, including discussion of potential uncertainty in hydrologic model parameters.")* and the reviewers lacked some important background material. I think the comments are very important for the interpretation of the results and are not sufficiently elucidated and not transparently addressed. And they might be related.

The added short description and the new figure with some example time series of SWE are both not sufficient. In particular, it is not proven whether so-called snow towers (e.g. see Freudiger et al. 2017 in WIRES Water for a review) build up only in the accumulation area of glaciers and nowhere else (in the past but in particular also in the future runs). The manuscript needs to be more informative and transparent about spatial and long-term cumulative, quantitative, information about cryosphere water balance terms.

*Response A: We have added a new figure (Figure 4 – also included here) that includes a snow tower metric to investigate the build-up of snow across melt seasons (i.e. "snow towers") in our study. The snow of glacier-free cells does not melt completely over summer and autumn but neither does it show a positive trend which suggests that the residual snow does not keep growing from year to year (plot b). We also added a snow tower statistic (plot a) that shows the combined area of the cells at the end of each hydrological year (31st of march) where SWE is greater than the amount of snowfall during the corresponding hydrological year. Only very few cells show such snow towers but it is interesting to see that the Tindex model produces more snow towers than the Anderson model. The effect on the water balance of these snow towers is rather small (< 2.5% of the entire area is potentially affected) but it would be interesting to use the latest WaSiM version that comes with a snow redistribution routine as part of future work. We have added the following material to the manuscript.*

*P6 L10 – P6 L21: "Due to the high annual precipitation totals the grid cells in the upper catchment could potentially be prone to large build-ups of snow that persist throughout the melting period and into the next hydrological year. As discussed by Freudiger et al. (2017) such snow towers can cause substantial errors in the modelled water balance and should therefore be made transparent. As indicated in Figure 4b the snow pack of some cells persists across all of the melt seasons with SWE values at the end of the melt season ranging between 7 and 46 mm (21 and 77 mm) for WaSiM-Anderson (WaSiM-Tindex). To further investigate the significance of this potential issue, all cells where the SWE value at the end of the hydrological year exceeded the annual snowfall (excluding glacier cells), were marked. The combined relative area of these cells is shown in Figure 4a and does not exceed 1% for WaSiM-Anderson, while for WaSiM-Tindex the affected area does not exceed 2% except for the years 1997 and 1998. The mean exceedance (SWE/snowfall) for the individual years based on all affected cells varies from 1.3 to 1.8 for WaSiM-Anderson and from 1.4 to 2.3 for WaSiM-Tindex. Thus, for a number of cells along the main divide the modelled build-ups of snow could be classified as potential snow towers. However, the combined area of these cells is small, which means that the effect on the modelled water balance is negligible."*

*P11 L32 – P12 L2: "Accounting for snow redistribution by avalanches or wind as described in Freudiger et al. (2017) should also be investigated as this would potentially reduce some of the relatively large accumulations of snow that were found to persist over multiple years on some cells in the upper catchment during the reference period."*

[Figure]

*Figure 4 (a) relative area of the Kawarau catchment where SWE exceeds annual snowfall for a hydrological year (e.g. 1.4.1992 – 31.3.1993) as an indicator for unrealistic build-up of snow (i.e. snow towers). (b) Modelled mean monthly snow water equivalent (SWE) based on the Tindex and Anderson simulations. (c) Observed and modelled (WaSiM-Anderson and WaSiM-Tindex) monthly runoff at Chards Rd for the reference period (1992-2012).*

*We have also added the following figure to the manuscript which shows the mean SWE storage for the six months with historically the greatest snow storage across the 21$^{st}$ century.*

*P10 L7 – P10 L12: "As shown (Figure 11) by the transient simulations of mean SWE from July to December (months with historically the highest SWE) there is no clear distinction between the A1B and A2 simulations until the last quarter of the 21$^{st}$ century when the median of the A2 runs stays consistently under the A1B runs. While the median SWE based on all 16 QM runs decreases substantially (60%) during the 110-year period, years with large accumulations of SWE still occur in the second half of the 21$^{st}$ century for individual members and years. The negative trend was found to be more pronounced for the glaciers in the catchment, which are projected to lose 93% of their volume (91% based on A1B and 94% based on A2) by 2099."*

[Figure]

*Figure 11 The mean SWE of the six months with the historically highest SWE (July to December) is shown for the 16 transient QM simulations (including QM-Anderson and QM-Tindex).*

The main aim is climate change assessment and hence long-term changes. Hence, this long term change needs to be the benchmark for the model simulation in the past. In such an environment long-term accumulation of snow and the correct glacier evolution (such as retreat) are interdependent and will be primary determinants of long-term changes to the water balance (and thus changes in the hydropower potential which is cited as the underlying motivation).

Indeed, now after learning from the revised text that there is considerable glacier area in the modelled catchment, I need to request more modelling details a) how snow redistribution is handled in WaSIM and b) how the glacier(s?) and glacier change is/are treated within the model. Please also add exactly how much the glacierized area is in the here-shown modelled catchment, not the entire Clutha (as the new sentence says now) and in particular how much the streamflow contribution from glacier mass balance changes is. A correct representation of the long-term glacier change will be the best proof for a generally correct snow model as well!

*Response B:* We have added the following sentences to the manuscript to clarify how much ice volume is estimated to be found inside the Kawarau catchment (P4 L4 – P4 L9): *"The ice-covered area inside the Kawarau catchment is small and based on 2001/2002 satellite imagery amounts to 1.8% or 84 km2 (New Zealand's Land Cover Database v3.0 as published by Landcare Research in 2012). Chinn (2001) estimated the volume of ice in the Kawarau basin at 2.25 km3 using the area mean-depth relationship of the World Glacier Monitoring Service. Thus, based on Chinn's (2001) estimate the entire water stored in the glaciers of the Kawarau could only sustain its mean flow for ~123 days, which highlights that glaciers in the catchment are less important from a hydrological perspective."*

The reviewers pointed out unexplained discrepancies in the observed and modelled hydrograph dynamics. The sentence on parameter uncertainly added is not a sufficient address to that concern.

*Response C: Although we tried to account for as many uncertainty sources as possible in this study, the sources of uncertainty that were included are GCM, emission scenario, bias correction and snow model. Hence parameter uncertainty was not accounted for and can/could only be investigated as part of future work. At this point we could only speculate about the role of parameter uncertainty and therefore decided that this would go beyond the scope of this paper. The discrepancies in the hydrographs as related to inaccuracies in the modelled seasonal snow storage are already described and discussed in detail in the manuscript (P11 L11 – P12 L7).*

May these differences and somewhat systematic errors be due also to the glacier melt modelling?

*Response D: Very unlikely because the change in glacier volume is small between individual years when compared to the amplitude of the seasonal snow storage.*

The runoff generation not well represented needs to be better assessed and clearly explained and discussed - or transparently ruled out.

*Response E: There is no direct validation of runoff processes or similar due to the lack of such records/data sets. It is however very likely that the bias in the flow regime is due to the snow model as discussed in the text in quotation marks under Response C.*

For the SWE-comparison of the two model variants, it would be necessary to be more transparent on how snow accumulation and melt are treated also within the glacier mass balance model component (normally, models convert snow to firn and hence 'extract' snow on glacier but not off-glacier from the water balance, snow may be differently redistributed or melt differently parameterized on or off-glacier ,etc.... - there are many potential differences that may affect the entire budget and hence the catchment-wide comparison of SWE and runoff).

*Response F: As stated earlier glaciers are not very important in contributing to the generation of flow. We have added some additional information around how the glacier model works.*

*P4 L17 – P4 L19: "The glacier model uses three degree-day factors to calculate melt from the three storage components snow, firn and ice. After each year the remaining snow of a cell forms a new firn layer and once the firn stack reaches seven layers (as recommended by Schulla (2012)) the lowest firn layer turns into ice."*

*P5 L16 – P5 L20: "Opposed to the other sub-models the glacier model was calibrated manually for the entire Clutha catchment using the annual volume estimates (1994-2010) of Willsman (2011) as calibration (1994-2001) and validation (2002-2010) source. The degree day factor (DDF) of ice (7.17 mm°C-1d-1) was based on the study of Anderson et al. (2006), while the DDF of snow (3.80 mm°C-1d-*

1) was calibrated manually. The DDF of firn (5.54 mm°C-1d-1) was calculated by averaging the DDFs of snow and ice, to ensure a physically sound relationship between the three parameters (DDF of ice > DDF of firn > DDF of snow)."

If the ice melt contribution to generated runoff is too small to be relevant, the following comments may perhaps be disregarded. But if glacier ice melt is a substantial contribution (which the reader doesn't learn in the current manuscript), the necessary information on modelling must be added to the manuscript to make the study results credible and usable.

*Response G: Thank you for those detailed comments around the modelling of permanent snow and ice. We have made some changes to the manuscript to avoid confusion around the estimated volume of glaciers inside the Kawarau catchment (Response B). The catchment contains 55% of the Clutha's ice cover but the overall volume is rather small in the Kawarau as well as in the other headwater sub-catchments of the Clutha. Based on the volume estimate (2.25 km3) of Chinn (2001), evenly distributed over the Kawarau catchment the ice volume corresponds to 496 mm which is small compared to the annual catchment precipitation of 2007 mm. The water stored in the glaciers of the Kawarau could only sustain its mean flow for ~123 days.*

The Reviewers criticized the lack of description of the calibration procedure and the quality of the reference period simulation against which to assess future changes. Here, the dissertation by Jobst 2017 is cited, but if these details are relevant, and they are, and as this is not peer reviewed literature, the details need to be described here instead. Otherwise, it cannot be evaluated whether the quality of the model allows the conclusions drawn. Please include the calibration strategy and multi-criteria data information as it is necessary to assess how well the correct runoff generation processes are described and runoff is not generated for the wrong reason. As we all know, it is easy to model very seasonal glacial hydrographs because of the possible compensation of accumulation and melt by parameters of the hydrological model and NS efficiencies have limited value (Schaefli and Gupta, HP - Do Nash Values have value?).

*Response H:  We understand that the implementation of the hydrological model is a fundamental part of accepting model results. Hence, we have tried to make the implementation and calibration process that was used to set up WaSiM clearer. Unfortunately, no data that would have allowed for the direct validation of runoff generation processes were available (e.g. groundwater table, SWE records or soil moisture) and therefore this could not be tested. We would like to stress that this is not a glacial hydrograph and it is very unlikely that glacier processes that misrepresented in the model overshadow the effect of the snow melt routine. The new figure added shows the calibration workflow that was used to calibrate the model and the calibrated parameters are explained in the figure's caption and in the added section (see below.)*

*P5 L5 – P5 L20: "First the parameters controlling subsurface flow were calibrated using the Nash-Sutcliffe model coefficient of efficiency (NSE). Therefore, the NSE was based on daily logarithmic streamflow values (NSElog) to account for the physical link between low flow conditions and subsurface flow components of WaSiM. The remaining parameters, controlling surface flow were calibrated using regular daily streamflow values (standard NSE) to preserve the sensitivity of the NSE to flood peaks. A regionalisation based on spatial proximity and topographical similarity was carried out next (see map in Figure 2) to parameterise sub-catchments that were ungauged or only had short records of streamflow (e.g. parameters of sub-catchment 4 → sub-catchment 3).*

*In the following step the two snow models were calibrated for three separate headwater sub-catchments (gauges: The Hillocks, Peat's Hut and West Wanaka as shown in Figure 1b) against*

*monthly streamflow (NSEmo) with the Nash-Sutcliffe criterion of efficiency (NSE) as the objective function. The resulting parameter sets were then averaged resulting in a global parameter set for each of the two snow models respectively (Table 1).*

*Opposed to the other sub-models the glacier model was calibrated manually for the entire Clutha catchment using the annual volume estimates (1994-2010) of Willsman (2011) as calibration (1994-2001) and validation (2002-2010) source. The degree day factor (DDF) of ice (7.17 mm°C-1d-1) was based on the study of Anderson et al. (2006), while the DDF of snow (3.80 mm°C-1d-1) was calibrated manually. The DDF of firn (5.54 mm°C-1d-1) was calculated by averaging the DDFs of snow and ice, to ensure a physically sound relationship between the three parameters (DDF of ice > DDF of firn > DDF of snow)."*

[Figure]

*Figure 2 The calibration workflow that was used to calibrate WaSiM. For each calibration step (either manual or automatic) information is provided as follows: "calibrated parameters | calibrated sub-catchments | performance criterium used". The parameters controlling subsurface flow encompass the drainage density for interflow (dr), the storage coefficient of interflow (kI) and the groundwater conductivity in x or y-direction. Surface flow is controlled by the storage coefficient of surface runoff (kD) and the fraction of snowmelt directly becoming surface runoff (QDsnow). The parameters of the snow models are shown in Table 1.*

Necessary information to be added also includes what happened to the glaciers between 1992 and 2012. What does that change entail for the calibrated parameters and resulting over-/underestimations in the validation period (see reviewer comments)?

*Response I: We have provided some more information on the change in the glacier volume during the reference period. The decrease from 540 mm to 265 mm during the reference period is substantial but when compared to the ice volume estimates of Willsman (2011) the model results can be considered as reasonable. Willsman (2011) found a decrease of 29% for the entire Clutha catchment between 1994 and 2010 which is less than the decrease found in this study for the Kawarau (42%). Unfortunately, there are no volume estimates for the Kawarau catchment but as shown in the figure below, which is taken from Jobst (2017) and not included in this article (number of figures is already relatively high), for the entire Clutha the modelled ice volume agrees well with the estimates of Willsman (2011) in 2010.*

*P6 L6 – P6 L9: "The modelled glacier storage decreased from 540 mm to 265 mm during the reference period. For comparison Willsman (2011) found a decrease of 29% for the entire Clutha catchment between 1994 and 2010, which is less than the decrease modelled here for the Kawarau (42%). As there are no other studies that have quantified the glacier volume of the Kawarau the results of this study couldn't be assessed any further."*

[Figure]

Figure 3.17 Modelled glacier volume (Volume$_{MOD}$) inside the Clutha catchment versus the volume estimates of Willsman (2011) (Volume$_{OBS}$). The modelled glacierized area (Area$_{MOD}$) is also displayed.

Was this change also the reason for the (to me very unusual) choice of a short recent calibration period and long past validation period? From my experience in such environments long model spin ups are required and calibration over phases of wet/cold and warm/dry phases necessary. Short calibration periods may lead to biased storage fillings and thus biased results. The motivation and procedure of these choices also need to be more transparently described as well - reader cannot be referred to read a thesis to learn this important methodological reasoning.

*Response J: As stated on P4 L29-31 "The last four hydrological years of the reference period were chosen for calibration because of the higher density of weather stations compared to previous years and a better consistency of the streamflow records." The rather short calibration period was also selected to reduce the otherwise too extensive processing time of WaSiM at the relatively high spatial resolution of 1 km². Although this can be regarded as a nonstandard approach the long validation period allowed for a robust assessment of the model's performance. Hence the following explanation has been added to the manuscript.*

*P4 L 31 – P5 L1: "Further the relatively short calibration period constitutes a compromise between a reasonable processing time and a sufficient number of iterations (as part of the auto-calibration). Although the calibration period is rather short this means that the longer validation period allows for a robust assessment of the hydrological model's performance.*

*After a two-year model spin-up the individual sub-models of WaSiM…"*

In addition to the question how well was the long-term glacier change in the past was modelled (see former comment that this is the best evaluation of a good snow model), it is also important to learn what happened to the glacier in the future simulations as the loss or gain will be reflected in the water balance. A future hydrograph without that information is not useful.

*Response K: As mentioned above the ice volume has decreased from 540 mm to 265 mm during the reference period. It should be noted that the average annual ice volume lost is small (i.e. 14 mm) compared to the annual amplitude of the snow storage which based on WaSiM-Anderson ranges between 33 in autumn and 286 mm in spring (i.e. 253 mm).*

In summary, the manuscript does not yet go far enough into modelling reasons and therefore my expectation is that it will also not have the impact it perhaps deserves. The conclusions need to go beyond the general warmer-less snow paradigm to make an interesting contribution to international science in the field. Therefore, I need to ask for further revisions regarding more transparency of the modelling details and more in-depth analyses of the reasons for the difference and range in changes founds.

*Response L: The main motivation of this study was to investigate some of the key uncertainty sources in hydrological climate change impact studies for a representative headwater sub-catchment of the Clutha River (New Zealand's largest catchment). The role of the snow model was also investigated in detail and the findings are important for both alpine catchments in general and for the relatively unexplored domain of the Southern Alps. As WaSiM is a widely used hydrological model this study makes an important contribution to the existing studies that have successfully used WaSiM as a climate change impact tool.*

*We would like to stress the point that both reviewers had suggested minor changes but were pleased with the structure and the general content of the manuscript (Reviewer 1: "The manuscript is clearly written, and I enjoyed reading it" and Reviewer 2: "The paper is well written and comprehensive"). At this stage we believe that we have addressed the comments of the reviewers to a satisfactory degree. Adding more material around the implementation of the model would make this paper less concise (at this point the manuscript has already reached a relatively high word count of 7627). We also believe that the uncertainty analysis goes well beyond a simple "warmer, less snow and therefore more winter runoff" study. Changes in temperature and precipitation were the driving processes but we believe that the results shown here would be of interest to a wider community as this study generated some important insights around the role of the snow model and how the latter compares with the more commonly investigated uncertainty components.*

Further Editorial comments

Figure 1: Please make this figure more specific to this study! Legend for the red polygon is missing. What is this? Is it relevant for this study at all? Otherwise please delete this polygon. 'snow cal' is not a nice or useful abbreviation here - please write out what is meant with that in this introductory figure. The order of a, b, c is very confusing/unusual. Please switch graphs or a, b,c to go from upper left to lower right as is usually the custom. In fact, a) is really not needed and anyway very dark colors hard to see anything. Instead, it would be more useful to give more landcover details of the modelled catchment, such as glaciers, forest vs grassland, etc. Most of what is in the map c) is not modelled and investigated in the study anyway (reservoirs and lakes only mentioned once as overall motivation).

*Changed as suggested*

[Figure]

**Figure 1** *Maps showing (a) New Zealand with the Clutha catchment located in the lower South Island, (b) the sub-catchments of the gauges recording discharge used in this study (note that the West Wanaka sub-catchment is outside the Kawarau catchment) and (c) a land cover classification of the Kawarau sub-catchment based on New Zealand's Land Cover Database (LCDB v3.0) that was published by Landcare Research in 2012.*

Figures 7-9 Axis labels are too small and too close to the axes. There is enough space to make them considerably larger and keep a bit of distance from the axes.

*Changed as suggested*

Figure 10a is not possible to understand without any indication what is on the x axis - what are the different ranges? The colors repeat, so it is not at all clear to me (and likely other readers) what the difference among these bars is.

*Additional information has been added to the figure caption including a reference to a more detailed explanation of this figure (visualisation of uncertainty by model component) in section 3.2. This figure could potentially be omitted but considering that the radar charts are derived from this figure we would suggest not deleting it. As mentioned the figure is explained in detail in section 3.2 but essentially serves as a visual tool to portray the range of the 32 permutations for each model component.*

10b-d. I suggest that these are sufficiently different to warrant a separate Figure 11 and then larger panels. At this size, it is hard to see differences clearly. Also, perhpas since the provided figure resolution is not acceptable. I assume that will be much better in the final production, but still, I think these are too small.

*Changed as suggested*

Table 1: give units where applicable

*Changed as suggested*

Table 3: why is SWE given in km3 and not also in mm similar to all other terms. This is not only highly unusual, it also doesn't allow comparison. Please change.

*Changed as suggested*

[revised manuscript text omitted]